# Deep learning approach for screening neonatal cerebral lesions on ultrasound in China

Zhouqin Lin [1,2,16], Haoming Zhang [3,4,16], Xingxing Duan [5,16], Yan Bai[6,16], Jian Wang [3,7,8,9], Qianhong Liang[10], Jingran Zhou[1,2], Fusui Xie[1,2], Zhen Shentu[2], Ruobing Huang[3,4], Yayan Chen[11], Hongkui Yu[12], Zongjie Weng[13], Dong Ni [4,8,9,14] ✉, Lei Liu[1,2] ✉ & Luyao Zhou [1,2,15] ✉

Timely and accurate diagnosis of severe neonatal cerebral lesions is critical for preventing long-term neurological damage and addressing life-threatening conditions. Cranial ultrasound is the primary screening tool, but the process is time-consuming and reliant on operator's proficiency. In this study, a deep-learning powered neonatal cerebral lesions screening system capable of automatically extracting standard views from cranial ultrasound videos and identifying cases with severe cerebral lesions is developed based on 8,757 neonatal cranial ultrasound images. The system demonstrates an area under the curve of 0.982 and 0.944, with sensitivities of 0.875 and 0.962 on internal and external video datasets, respectively. Furthermore, the system outperforms junior radiologists and performs on par with mid-level radiologists, with 55.11% faster examination efficiency. In conclusion, the developed system can automatically extract standard views and make correct diagnosis with efficiency from cranial ultrasound videos and might be useful to deploy in multiple application scenarios.

Neonatal cerebral lesions refer to a range of neurological abnormalities, including intraventricular hemorrhage (IVH), periventricular leukomalacia (PVL), and ventriculomegaly. These conditions are particularly common in preterm and low birth weight (BW) infants, as well as neonates who experience hypoxia-ischemia during delivery[1,2].

Among them, IVH is the most common and extensively studied form of brain injury[3]. The severity of IVH is typically categorized into four grades using the Papile grading system, with grades III and IV considered severe[4,5]. Severe IVH often occurs alongside complications such as hydrocephalus and PVL, significantly increasing neonatal

[1]Department of Medical Ultrasonics, Shenzhen Children's Hospital, Shenzhen, PR China. [2]Shenzhen Pediatrics Institute of Shantou University Medical College, Shenzhen, PR China. [3]School of Biomedical Engineering, Shenzhen University, Shenzhen, PR China. [4]Medical Ultrasound Image Computing (MUSIC) Lab, School of Biomedical Engineering, Medical School, Shenzhen University, Shenzhen, PR China. [5]Department of Ultrasonography, Changsha Hospital for Maternal and Child Health Care, Changsha, PR China. [6]Sichuan Provincial Women's and Children's Hospital/The Affiliated Women's and Children's Hospital of Chengdu Medical College, Chengdu, PR China. [7]College of Computer Science and Software Engineering, Shenzhen University, Shenzhen, PR China. [8]School of Artificial Intelligence, Shenzhen University, Shenzhen, PR China. [9]National Engineering Laboratory for Big Data System Computing Technology, Shenzhen University, Shenzhen, PR China. [10]Panyu Maternal and Child Care Service Centre of Guangzhou, Guangzhou, PR China. [11]Ultrasound Department of Longhua District Maternal and Child Healthcare Hospital, Shenzhen, PR China. [12]Department of ultrasound, Shenzhen Baoan Women's and Children's Hospital, Shenzhen, PR China. [13]Department of Medical Ultrasonics, Fujian Maternity and Child Health Hospital, College of Clinical Medicine for Obstetrics & Gynecology and Pediatrics, Fujian Medical University, Fuzhou, PR China. [14]School of Biomedical Engineering and Informatics, Nanjing Medical University, Nanjing, PR China. [15]Medical School, Shenzhen University, Shenzhen, PR China. [16]These authors contributed equally: Zhouqin Lin, Haoming Zhang, Xingxing Duan, Yan Bai. ✉e-mail: nidong@szu.edu.cn; szliulei19@163.com; zhouly@email.szu.edu.cn

mortality and the risk of adverse neurodevelopmental outcomes[6]. For example, grade III–IV IVH is associated with nearly a five-fold increase in the risk of cerebral palsy (odds ratio: 4.98, 95% CI: 4.13–6.00)[7]. These conditions may also lead to long-term survivors experiencing intellectual disabilities, cognitive impairments, visual disturbances, or social issues[8]. Consequently, the timely identification and treatment of cerebral lesions in high-risk neonates are crucial for their healthy development.

Cranial ultrasound (CUS) is the primary modality for diagnosing and monitoring neonatal cerebral lesions due to its portability, non-invasive nature, and lower cost compared to MRI[9–11]. The standard anterior fontanelle approach in CUS provides clear visualization of critical brain regions, including the lateral ventricles, thalamus, and periventricular white matter—areas where severe cerebral lesions are commonly found[12,13]. The recommended CUS examinations for preterm infants or neonates suspected of having IVH should be conducted on days 1, 3, 7, 14, 28, and at term-equivalent age[14]. Moreover, repeating CUS around the 28th day of life can detect parenchymal abnormalities and ventriculomegaly and provide prognostic information for further management[15]. Conducting high-quality CUS examinations requires experienced radiologists who can not only perform thorough scans but also interpret and diagnose CUS images or videos accurately. Unfortunately, a global shortage of skilled radiologists limits access to reliable screening and examination. This shortage impacts high-income countries such as the United States and the United Kingdom[16] and presents even greater challenges in low- and middle-income regions, where the prevalence of severe brain injuries in infants is often higher[2,17]. Therefore, there is an urgent need for an automated, reliable, and scalable tool that can assist in identifying high-risk cases.

Recent advancements in deep learning (DL) offer a transformative opportunity to address these challenges. Studies have demonstrated that DL models can detect diseases from medical images, comparable to human experts[18,19]. Unlike human radiologists who rely on subjective experience, DL models analyze images by identifying deep and complex patterns, which allows for faster, more consistent, and objective diagnoses. This capability could help alleviate the shortage of trained radiologists. In recent years, the development of DL for fetal cranial imaging[20,21] has refined prenatal diagnosis, laying the foundation for early intervention in neonatal brain disorders. For instance, a recent study has focused on binary classification of CUS images (e.g., normal vs. abnormal) in very preterm infants[22], but the clinical value of such models is limited. This work overlooks the distinction between mild and severe abnormalities of neonatal cerebral lesions. Mild abnormalities, such as low-grade IVH and small cysts, often resolve spontaneously, whereas severe cerebral lesions require urgent attention due to their potential for long-term damage[23]. Therefore, a more detailed severity classification is essential to accurately identify and prioritize the most critical cases.

To address this gap, we proposed a neonatal cerebral lesions screening system (NCLS) to identify neonates at high risk for severe cerebral lesions. The system includes a view extraction module to detect key anatomical structures and extract standard views from CUS videos and a diagnostic module that integrates multiple views from the same neonate to predict lesion severity. These modules were trained and developed using an internal development set consisting of 8757 neonatal CUS images, retrospectively collected from a single hospital, corresponding to 1518 cases. To evaluate the performance of NCLS, we employed two distinct test datasets. The first test dataset was an internal video test set, comprising 199 cases prospectively collected from the same hospital. The second test dataset was an external video dataset, which included 356 cases prospectively collected from three other centers, enabling us to assess the clinical potential and generalizability of NCLS.

## Result

### Dataset and readers

As shown in Fig. 1a, we collected 1518 cases (gestational age (GA) 35.00 (95% Confident Interval, CI: 25.14–40.86) weeks; from January 2021 to December 2022) from Shenzhen Children's Hospital (SZCH) to form the development dataset, which was further split into training and test sets in an 8:2 ratio. Additionally, we collected 199 cases (GA 36.14 (95% CI: 27.23–41.00) weeks; from January 2023 to June 2023) from the same hospital to form the internal test dataset. The external test dataset was collected from three centers (Guangzhou Panyu District Maternal and Child Health Care Hospital, GZMCH; Sichuan Provincial Maternity and Child Health Care Hospital, SCMCH; Changsha Hospital for Maternal and Child Health Care, CSMCH) and included 356 cases (GA 37.00 (95% CI: 26.57–40.86) weeks; from October 2023 to April 2024). Each case contained four to six CUS standard views and two CUS videos. The CUS standard views included the anterior horn view (AHV), third ventricle view (TVV), and body view (BV) in the coronal plane, as well as the midsagittal view (MSV) and right parasagittal view (RPSV), and left parasagittal view (LPSV) in the sagittal plane. The CUS videos included both coronal and sagittal sweeping videos. Table 1 provides a summary of the demographic information of included infants, and Supplementary Fig. 1 shows the distribution of different diseases, GA groups, and probe types in the datasets. Supplementary Tables 1 and 2 provide more information regarding imaging details, including transducer frequency, manufacturer, model name, depth, and more.

As shown in Fig. 1b, the workflow of NCLS consisted of two stages. Stage 1 focused on the anatomical structure detection task, with the goal of extracting standard views from CUS videos based on the detection results. Stage 2 focused on the diagnostic task, aiming to classify whether the case has severe cerebral lesions based on the multiple standard views. We trained and validated our system using five-fold cross-validation in the development dataset and evaluated its performance in the internal and external test datasets. Moreover, twenty-five radiologists with varying levels of experience in CUS diagnosis were recruited for the study. Among them, nine junior radiologists had about 1–2 years of experience in clinic CUS diagnosis, eleven mid-level radiologists had 3–7 CUS diagnosis experience, and five senior radiologists had more than 8 years of CUS diagnosis experience, including two authoritative senior radiologists with over 15 years of experience. These five senior radiologists contributed to the annotation for the development dataset, and the two authoritative senior radiologists were responsible for annotating the internal and external test datasets. Note that each CUS image in the development set was independently diagnosed as normal, IVH, ependymal cyst, ventricular dilation, hydrocephalus, or PVL, based solely on the content presented in the image[9]. These individual image diagnoses were then combined to form a case diagnosis. In contrast, the diagnostic labels for cases in the internal and external test sets were determined based on CUS videos, which aligns with the clinical diagnostic workflow. Cases with grade III or IV IVH, PVL, or hydrocephalus were classified as severe abnormalities and labeled as 1[5,23,24]. Cases with grade I or II IVH, ependymal cysts, or minor ventricular dilation, along with normal cases, were categorized as mild abnormalities and labeled as 0.

### Comparison of NCLS and radiologists

To assess the diagnostic performance of the NCLS, we compared it with nine junior radiologists and eleven mid-level radiologists on the internal and external test datasets (Fig. 1c). Table 2 shows the comparison results. In the internal test dataset, the NCLS achieved a sensitivity of 0.875 (95% CI: 0.687–1.000), specificity of 0.934 (95% CI: 0.897–0.967), AUC of 0.982 (95% CI: 0.958–0.997), and an F1-score of 0.667. In comparison, the average performance of the junior radiologists was as follows: sensitivity of 0.875 (95% CI: 0.810–0.924), specificity of 0.851 (95% CI: 0.833–0.868), AUC of 0.863, and F1-score

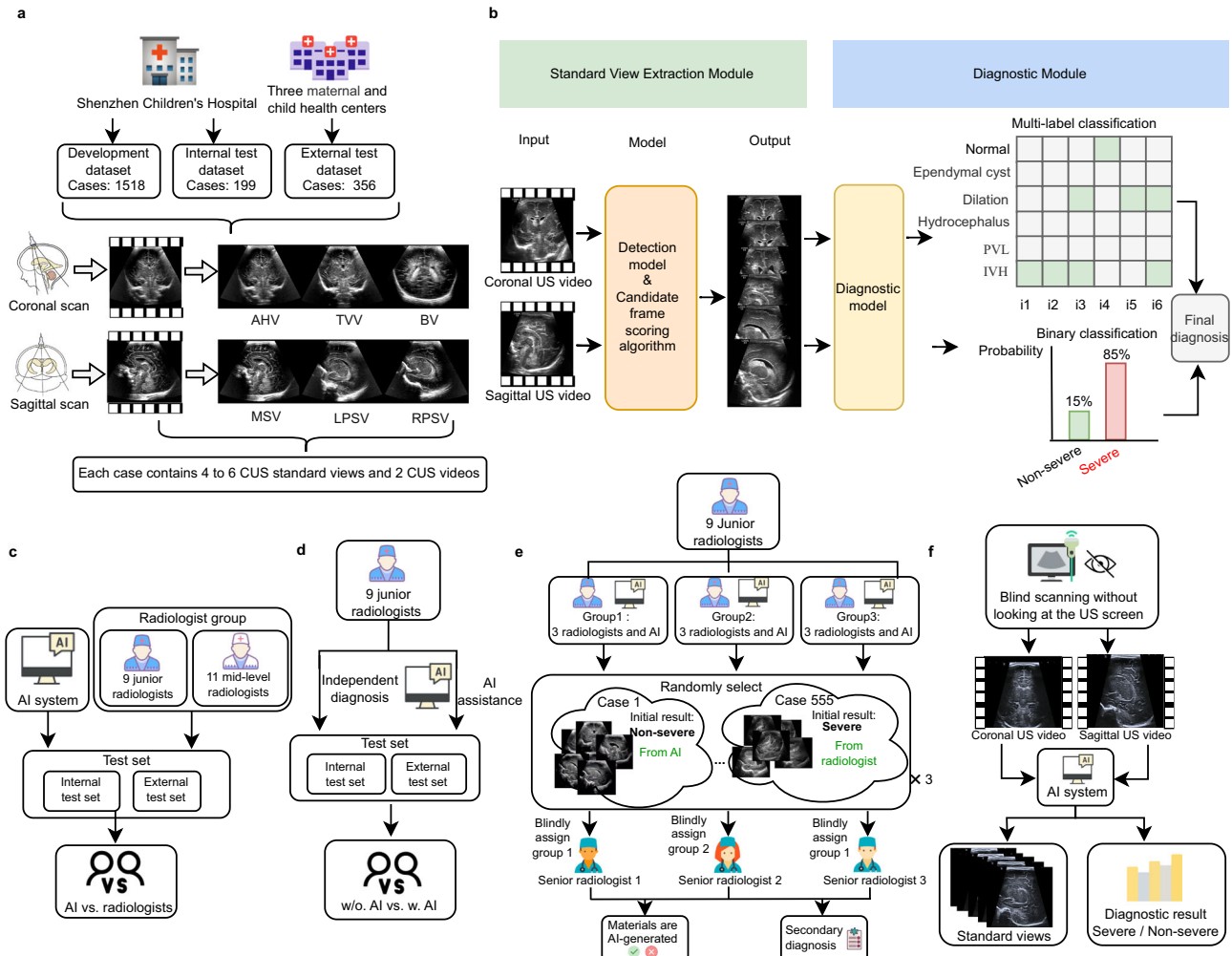

**Fig. 1 | Dataset, NCLS workflow, and study design. a** Dataset division and sources, as well as the images, videos, and scanning methods used to acquire imaging data for each case. **b** The workflow of the NCLS inference. The system inputs CUS videos and outputs extracted standard views and severity diagnosis results. **c** The workflow of the comparison of NCLS and radiologists. **d** Comparison of junior radiologists with and without AI assistance. **e** Blind and randomized trial of junior radiologists versus AI. **f** Evaluation of NCLS on blind-sweep data. NCLS, neonatal cerebral lesions screening. AHV anterior horn view, TVV third ventricle view, BV body view of the lateral ventricle, MSV midsagittal view, RPSV right parasagittal view, LPSV left parasagittal view. Icons by Icons8.

**Table 1 | Baseline characteristics of the development dataset, internal and external test datasets**

| Parameter | Development dataset | | | Internal test set | External test dataset | | |
|---|---|---|---|---|---|---|---|
| | Train | Test | P value | | SCMCH | CSMCH | GZMCH |
| Count | 1206 (79.4%) | 312 (20.6%) | – | 199 | 109 (30.6%) | 117 (32.8%) | 130 (36.5%) |
| Sex | | | | | | | |
| Male | 726 (60.2%) | 194 (62.2%) | 0.567 | 140 (70.3%) | 61 (56.0%) | 67 (57.3%) | 62 (47.7%) |
| Female | 480 (39.8%) | 118 (37.8%) | | 59 (29.7%) | 48 (44.0%) | 50 (42.7%) | 68 (52.3%) |
| GA (week) | 34.7 (25.1– 40.7) | 36.4 (26.9– 41.4) | 0.979 | 36.1 (27.2– 41.0) | 35.3 (24.9– 40.3) | 38.7 (27.3–40.6) | 36.7 (26.3– 41.0) |
| BW (gram) | 2200.0 (686.5– 4017.0) | 2420.0 (891.0– 4024.0) | 0.031 | 2505.0 (710.0– 4021.8) | 2280.0 (750.0– 3474.4) | 3000.0 (1050.0– 4050.0) | 2470.0 (743.5–3879.0) |
| Age (day) | 12.0 (1.0– 148.4) | 13.0 (1.0– 177.2) | 0.744 | 11.0 (1.0– 47.2) | 2.0 (1.0– 54.2) | 6.0 (1.0– 30.4) | 2.5 (1.0– 22.3) |
| Delivery | | | | | | | |
| CS | 665 (55.1%) | 174 (55.8%) | 0.937 | 122 (61.3%) | 79 (72.5%) | 45 (38.5%) | 66 (50.8%) |
| VD | 541 (44.9%) | 138 (44.2%) | | 77 (38.7%) | 30 (27.5%) | 72 (61.5%) | 64 (49.2%) |
| Apgar score | | | | | | | |
| 1 min | 9.0 (1.00– 10.0) | 9.0 (1.0– 10.0) | 0.764 | 9.0 (3.0– 10.0) | 10.0 (3.6– 10.0) | 10.0 (2.0– 10.0) | 9.0 (2.1– 10.0) |
| 5 min | 10.0 (4.0– 10.0) | 10.0 (2.6– 10.0) | 0.631 | 10.0 (5.0– 10.0) | 10.0 (8.0– 10.0) | 10.0 (7.0– 10.0) | 10.0 (5.0– 10.0) |
| 10 min | 10.0 (5.0– 10.0) | 10.0 (3.0– 10.0) | 0.716 | 10.0 (6.0– 10.0) | 10.0 (8.4– 10.0) | 5.0 (5.00– 7.00) | 10.0 (5.3– 10.0) |

For categorical variables, data are shown as (*N*, %); for non-normally distributed continuous variables, data are shown as (*N*, 95% CI). *P* values are two-tailed, calculated by the Chi-square or Mann–Whitney *U* test. Source data are provided as a Source data file. CS and VD refer to Cesarean and vaginal delivery; SCMCH, CSMCH, and GZMCH represent Sichuan, Changsha, and Guangzhou Maternal and Child Health Hospitals, respectively.
*GA* gestational age, *BW* birth weight.

**Table 2 | Comparison of NCLS and radiologists**

| Sets | Group | SEN | SPE | PPV | NPV | F1 | AUC | P value |
|---|---|---|---|---|---|---|---|---|
| Internal test set | NCLS | 87.50 (68.74–100.0) | 93.44 (89.73–96.70) | 53.85 (34.45–72.41) | 98.84 (97.02–100.0) | 66.67 (48.26–81.25) | 98.16 (95.89–99.65) | – |
| | Mid-level radiologists | 81.25 (74.69–86.73) | 98.66 (98.05–99.11) | 84.12 (77.74–89.26) | 98.37 (97.71–98.87) | 82.66 (78.09–86.85) | 89.95 (86.90–92.75) | 0.0009 |
| | Junior radiologists (w/o AI) | 87.50 (80.97–92.42) | 85.12 (83.31–86.81) | 33.96 (29.15–39.03) | 98.73 (98.00–99.25) | 48.93 (43.76–54.29) | 86.31 (83.09–88.91) | 0.0019 |
| | Junior radiologists (w. AI) | 97.22 (93.04–99.24) | 89.37 (87.79–90.82) | 44.44 (38.87–50.12) | 99.73 (99.31–99.93) | 62.95 (55.90–66.53) | 93.30 (91.67–94.69) | 0.0097* |
| External test set | NCLS | 96.15 (86.94–100.0) | 92.73 (89.88–95.30) | 51.02 (36.36–65.91) | 99.67 (98.99–100.0) | 66.67 (52.17–77.78) | 94.44 (89.42–97.31) | – |
| | Mid-level radiologists | 71.68 (66.08–76.83) | 97.55 (96.99–98.03) | 69.73 (64.13–74.93) | 97.76 (97.23–98.22) | 70.69 (66.67–74.60) | 84.61 (82.95–87.21) | 0.0004 |
| | Junior radiologists (w/o AI) | 71.79 (65.56–77.46) | 92.36 (91.34–93.29) | 42.53 (37.60–47.57) | 97.56 (97.02–98.18) | 53.42 (48.47–57.62) | 82.08 (79.22–85.04) | 0.0019 |
| | Junior radiologists (w. AI) | 91.03 (86.61–94.36) | 91.25 (90.17–92.24) | 45.03 (40.49–49.64) | 99.23 (98.83–99.52) | 62.20 (55.67–64.35) | 91.14 (89.17–92.86) | 0.0039* |

All metrics are presented as percentages (95% CI). Values for the junior- and mid-level-radiologist groups represent the mean across individual readers. The junior radiologist group was further divided into two subgroups: with AI assistance and without AI assistance. Statistical significance was evaluated solely with a paired, one-sided Wilcoxon signed-rank test on the reader-level ΔAUC. P values without * indicate the comparison between the NCLS model and each radiologist group. P values marked with * refer to the comparison between junior radiologists with and without AI assistance. No adjustment for multiple comparisons was applied. Source data are provided as a Source data file. SEN sensitivity, SPE specificity, PPV positive predictive value, NPV negative predictive value, F1 F1-score.

of 0.489. The Fleiss' Kappa value was 0.3005 (*P* value < 0.001), indicating low agreement among the junior radiologists. For the mid-level radiologists, the average sensitivity was 0.813 (95% CI: 0.747–0.867), specificity was 0.986 (95% CI: 0.980–0.991), AUC was 0.900, and F1-score was 0.827. The Fleiss' Kappa value was 0.7615 (*P* value < 0.001), indicating strong agreement within the group. Figure 2a displays the AUC for each radiologist and the ROC curve of the NCLS. The sensitivity of NCLS was no less than that of the radiologist group, showing a comparable diagnostic performance to that of radiologists in severe cases. The specificity of the junior radiologists was significantly lower than that of the NCLS, indicating that junior radiologists tended to diagnose an excessive number of false-positive cases, which could lead to unnecessary additional work and increased patient burden. In terms of AUC and F1-score, the diagnostic performance of the NCLS overall surpassed that of the junior radiologists.

On the external test set, the NCLS achieved a sensitivity of 0.962 (95% CI: 0.869–1.000), specificity of 0.927 (95% CI: 0.899–0.953), AUC of 0.944 (95% CI: 0.894–0.973), and an F1-score of 0.667. In comparison, the average performance of the junior radiologists included a sensitivity of 0.718 (95% CI: 0.656–0.775), specificity of 0.924 (95% CI: 0.913–0.933), AUC of 0.821 (95% CI: 0.792–0.850), and an F1-score of 0.534. The Fleiss' Kappa value was 0.3950 (*P* value < 0.001), indicating low agreement within the group. For the mid-level radiologists, the average sensitivity was 0.717 (95% CI: 0.661–0.768), specificity was 0.976 (95% CI: 0.970–0.980), AUC was 0.846 (95% CI: 0.830–0.872), and the F1-score was 0.707. The Fleiss' Kappa value was 0.6072 (*P* value < 0.001), indicating strong agreement within the group. The NCLS showed significantly higher sensitivity than both the junior and mid-level radiologists, while maintaining a high level of specificity. Moreover, while the performance of the radiologists varied considerably between the two datasets, the NCLS maintained consistently high diagnostic performance and reliability. Performance of each radiologist was shown in Supplementary Tables 3–5. For diagnostic speed (including extracting standard views and diagnosis), NCLS took an average of 28.97 s per case, junior radiologists took an average of 60.13 s per case, and mid-level radiologists took an average of 63.42 s per case. The majority of time spent by NCLS was on detecting the anatomical structures in each frame of the video, whereas the number of frames in CUS videos was uncertain.

We further analyzed cases of missed diagnoses and false positives, as shown in Fig. 2b. The NCLS missed four cases, all of which were PVL. This can likely be attributed to the data imbalance, as PVL cases represented only 1.5% of the training set. Figure 3 shows the standard views extracted by the system on misdiagnosed cases. These views reveal the cystic white matter injury, allowing radiologists to easily identify the abnormalities. Moreover, the majority of false positives occurred in grade I or II IVH cases, likely due to their visual similarity to grade III IVH in imaging. For the radiologists, the distribution of diagnostic errors was more scattered compared to the NCLS, indicating greater variability among radiologists. Figure 4 shows the heatmaps generated from multiple views using Finer-CAM[25], and it reveals that the NCLS focuses on areas around the anterior horn and body of the lateral ventricles. Supplementary Fig. 2 shows the regions of interest (ROIs) annotated by the radiologist and the heatmaps, demonstrating a high degree of overlap between the areas of focus identified by the NCLS and those identified by the radiologist. We also assessed the classification performance of the NCLS for both binary classification (normal vs. abnormal) and three-tier classification (normal vs. mild vs. severe) across all test sets, with detailed information provided in Supplementary Tables 6–8. Additionally, we conducted subgroup analyses based on probe types and different GA groups, and the corresponding results can be found in Supplementary Tables 9 and 10. Furthermore, we also conducted an analysis based on the Papile and Volpe grading systems for IVH. Supplementary Fig. 3 shows the distribution changes of the original

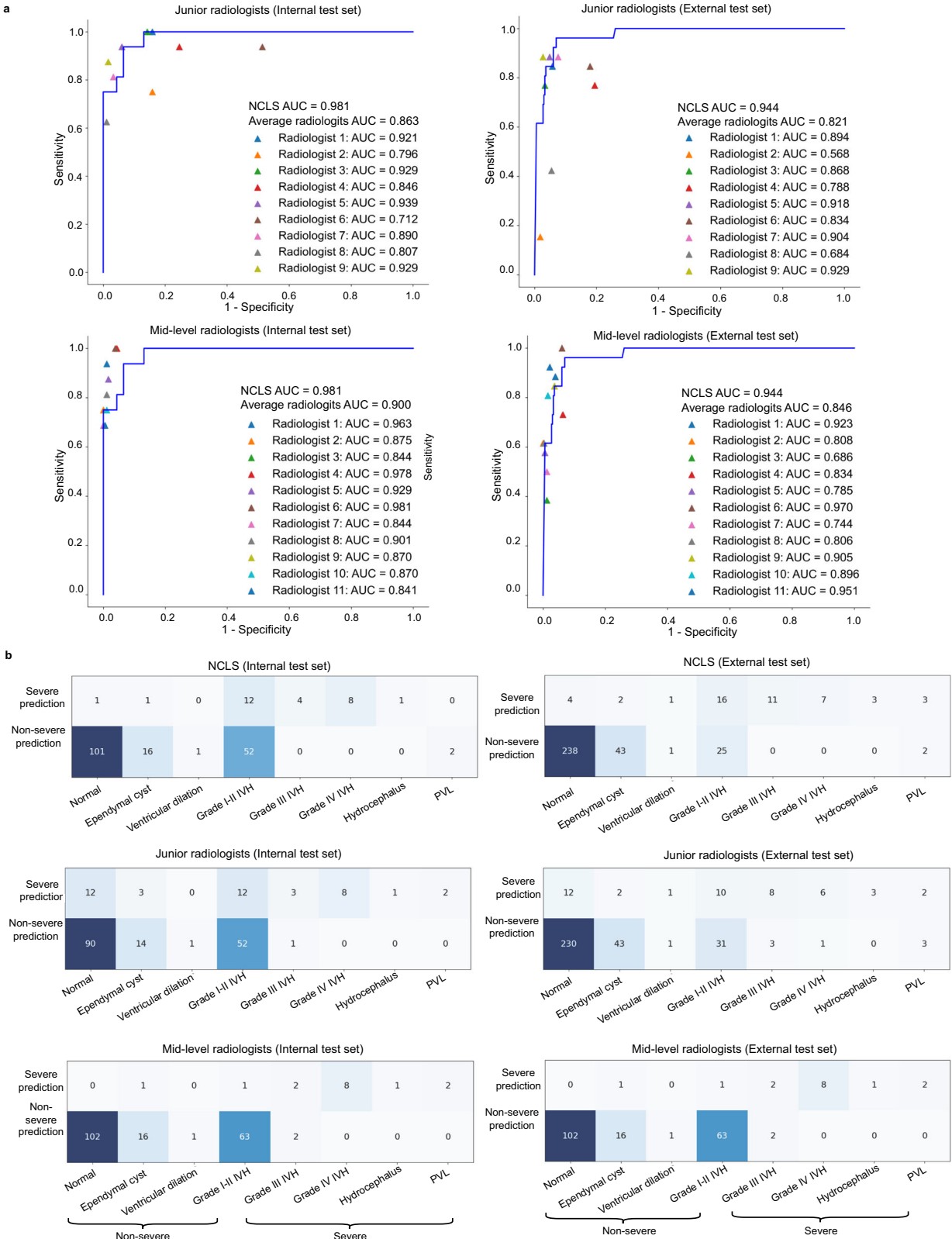

**Fig. 2 | Performance of the NCLS and radiologists. a** ROC curve of the NCLS and the AUC of each radiologist. **b** The diagnostic performance of NCLS and radiologists on specific conditions. For cases with normal, ependymal cyst, ventricular dilation, and grade I or II IVH, predicting the outcome as non-severe is considered a correct prediction. For other conditions, the reverse applies. The left column corresponds to the internal test set, while the right column corresponds to the external test set. In the confusion matrix, the specific numbers represent the count of the diagnostic outcomes of the model or radiologists for a specific condition. Source data are provided as a Source data file. AUC area of curve, NCLS neonatal cerebral lesions screening, PVL periventricular leukomalacia.

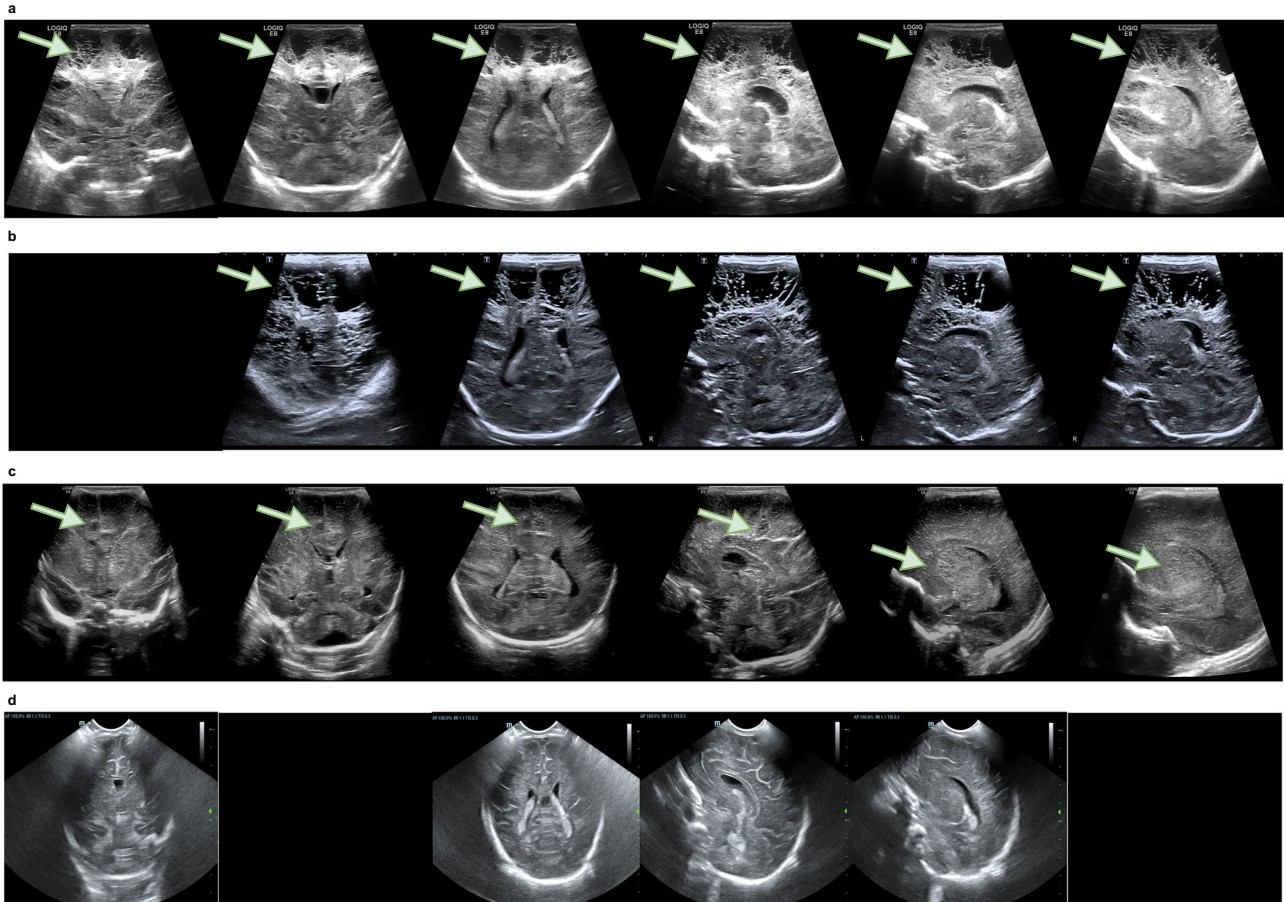

**Fig. 3 | Extracted standard views of misdiagnosed cases in the internal and external test sets. a–d** All four cases were PVL, with red arrows highlighting the presence of cystic periventricular white matter. The images that appeared entirely black indicate the absence of standard views. In (**d**), the cystic periventricular white matter was not discernible due to the lesion being located between frames of varying standard views, compounded by its small size.

Papile graded IVH data after being reclassified according to the Volpe grading system. Supplementary Table 11 provides the evaluation results of the NCLS using both grading systems. Additionally, Supplementary Fig. 4 illustrates the prognosis of IVH cases under both grading systems.

**Performance of junior radiologists with AI assistance**

From the analysis in the previous section, we concluded that the NCLS outperformed the junior radiologist group. We further evaluated the role of the system in assisting in diagnostic decision-making. Specifically, the junior radiologists were instructed to re-evaluate the internal and external sets with the assistance of the NCLS 1 month after their initial evaluation. During this process, the NCLS provided not only the diagnosis result but also standard views of each case (Fig. 1d). Table 2 presents the diagnostic results of junior radiologists with and without AI assistance. With AI assistance, in the internal test set, the average sensitivity of the junior radiologist group improved by 9.72% (95% CI: 2.08%–18.06%). Additionally, the average specificity improved by 4.25% (95% CI: −5.35% to 13.30%). In the external test set, the average sensitivity improved by 19.24% (95% CI: 3.85%–35.05%), and the average specificity showed a mean difference of −1.11% (95% CI: −5.87% to 4.24%). The result demonstrated that junior radiologists could significantly improve the accuracy of diagnosing severe cases with AI assistance, without compromising specificity. Figure 5a, b illustrates the changes in the metric of each radiologist. We observed that for seven out of nine junior radiologists in the internal test set and all junior radiologists in the external test set, sensitivity has improved, and the AUC values of all junior radiologists have

increased as well. These findings indicated that our system could effectively assist radiologists in making more efficient and reliable diagnoses.

We further analyzed the impact of AI assistance on junior radiologists. A total of 91.18% (62 out of 68) of the cases initially misdiagnosed by junior radiologists were corrected. Among 17 cases where junior radiologists made correct independent diagnoses but NCLS provided incorrect diagnoses, none were modified. Before incorporating AI assistance, a total of nine junior radiologists misdiagnosed 33 cases of grade III IVH and 24 cases of PVL. Additionally, six radiologists misdiagnosed 17 cases of grade IV IVH, and one radiologist misdiagnosed 3 cases of hydrocephalus. With AI assistance, only two junior radiologists misdiagnosed 2 cases of grade III IVH, two radiologists misdiagnosed 2 cases of grade IV IVH, one radiologist misdiagnosed 1 case of hydrocephalus, and nine radiologists misdiagnosed 20 cases of PVL. These results indicate that, with the assistance of our system, the diagnostic abilities of junior radiologists in severe IVH cases were improved, though the support provided in diagnosing PVL cases remains limited. Figure 5c shows the number of severe and non-severe diagnoses made by the junior radiologists with AI assistance for each specific condition. Figure 5d shows the Cohen's kappa values and Fleiss' kappa value within the group. The Fleiss' kappa value of the junior radiologist group with AI assistance was 0.7034 and 0.7526 on two test sets (P value < 0.001), showing a significant improvement compared to the Fleiss' kappa value of the junior radiologist group without AI assistance, indicating that AI assistance enhanced inter-rater agreement among the junior radiologists.

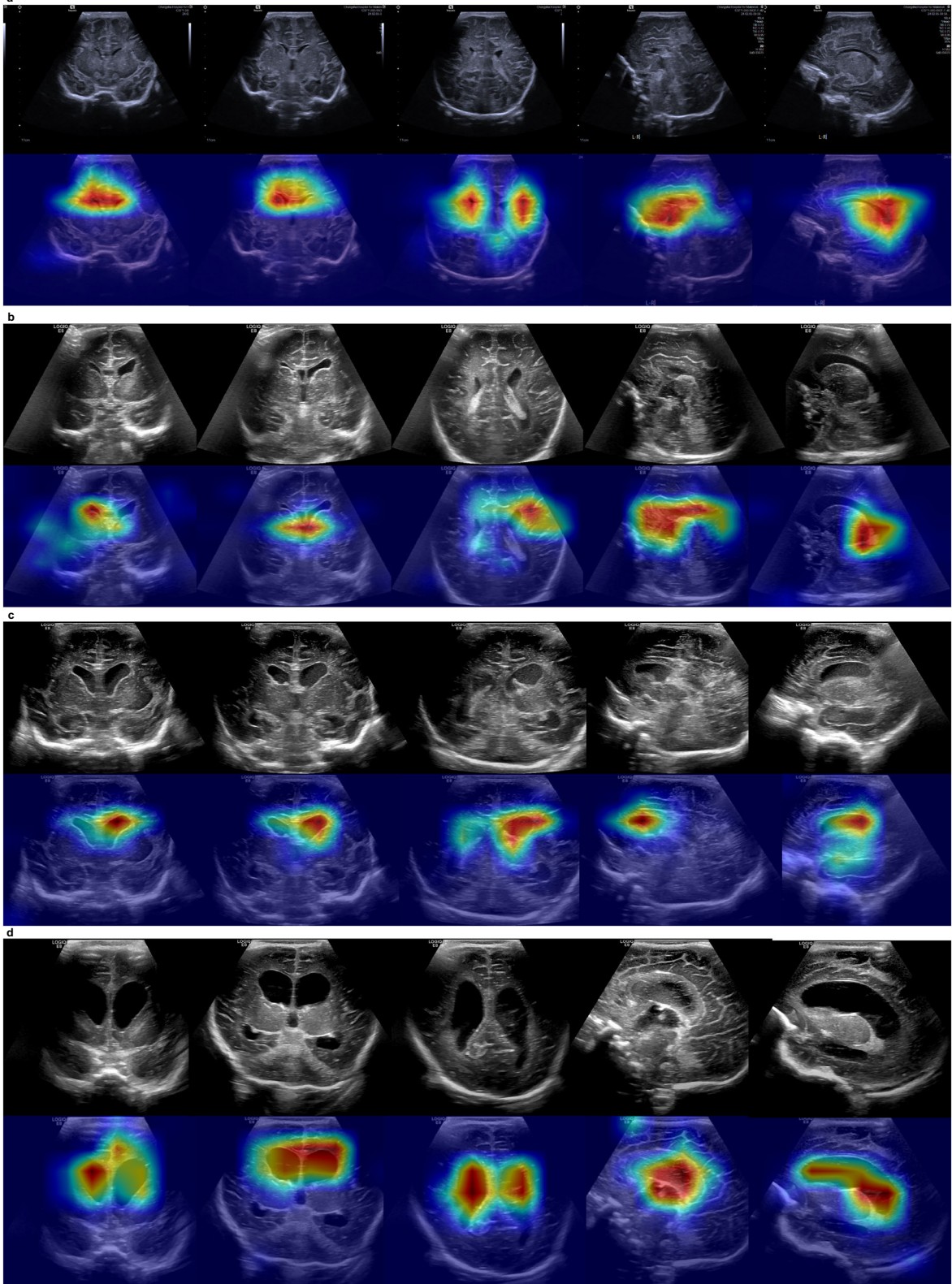

**Fig. 4 | Heatmap for certain cases. a** Normal cases. **b** Case with ventricle dilatation. **c**, **d** Cases with hydrocephalus and grade III IVH. The highlighted areas (red and yellow) represent the regions of significant attention by the model.

## Blind, randomized trial of junior radiologists versus AI

In order to evaluate the impact of the NCLS on the diagnostic decisions of radiologists and to objectively assess its diagnostic accuracy while minimizing potential biases introduced by the system, we implemented a blind and randomized trial. A total of nine junior radiologists were randomly and evenly categorized into three groups, and each group was assigned to one of three senior radiologists. For each case in both the internal and external test datasets, diagnostic materials were randomly selected by either a junior radiologist or the NCLS and presented to the senior radiologists under blind review. The senior

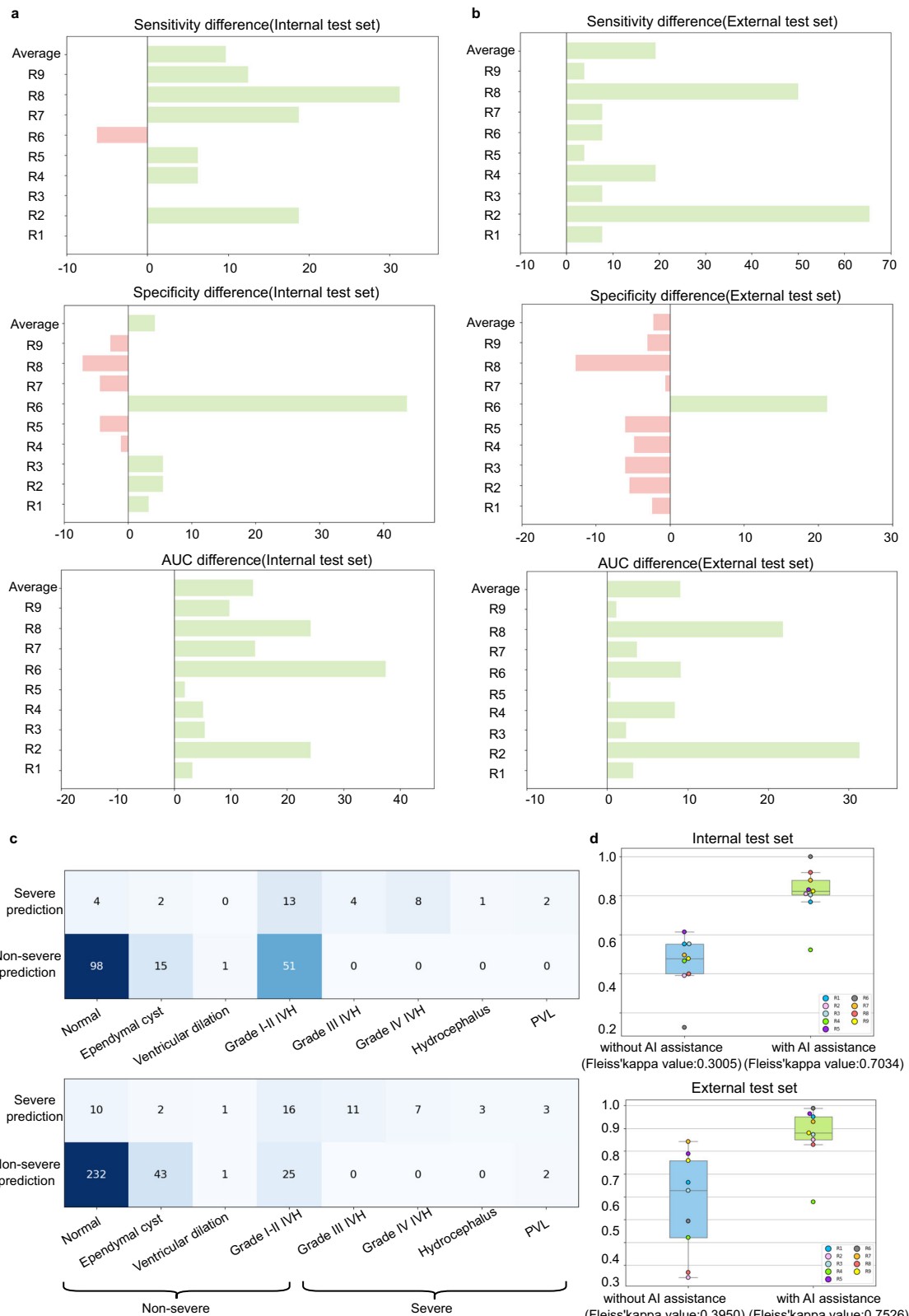

**Fig. 5 | Comparison of junior radiologists without and with AI assistance.**
**a**, **b** Improvement of metrics (sensitivity, specificity, AUC) with AI assistance on internal test set and external test set. Green color means improvement, and red color means decline. **c** The diagnostic performance of junior radiologists with AI assistance on specific conditions. **d** The consistency of junior radiologists without and with AI assistance. Box plots show Cohen's kappa values between AI and junior radiologists (*n* = 9 radiologists, biological replicates). Box plots display median (center line), 25th–75th percentiles (box bounds), whiskers extending to the minimum and maximum values within 1.5 × IQR from the quartiles, and individual data points overlaid. Each colored point represents one physician's diagnostic agreement with AI. The left column represents consistency without AI assistance, while the right column represents consistency with AI assistance. Source data are provided as a Source data file. R radiologist, PVL periventricular leukomalacia.

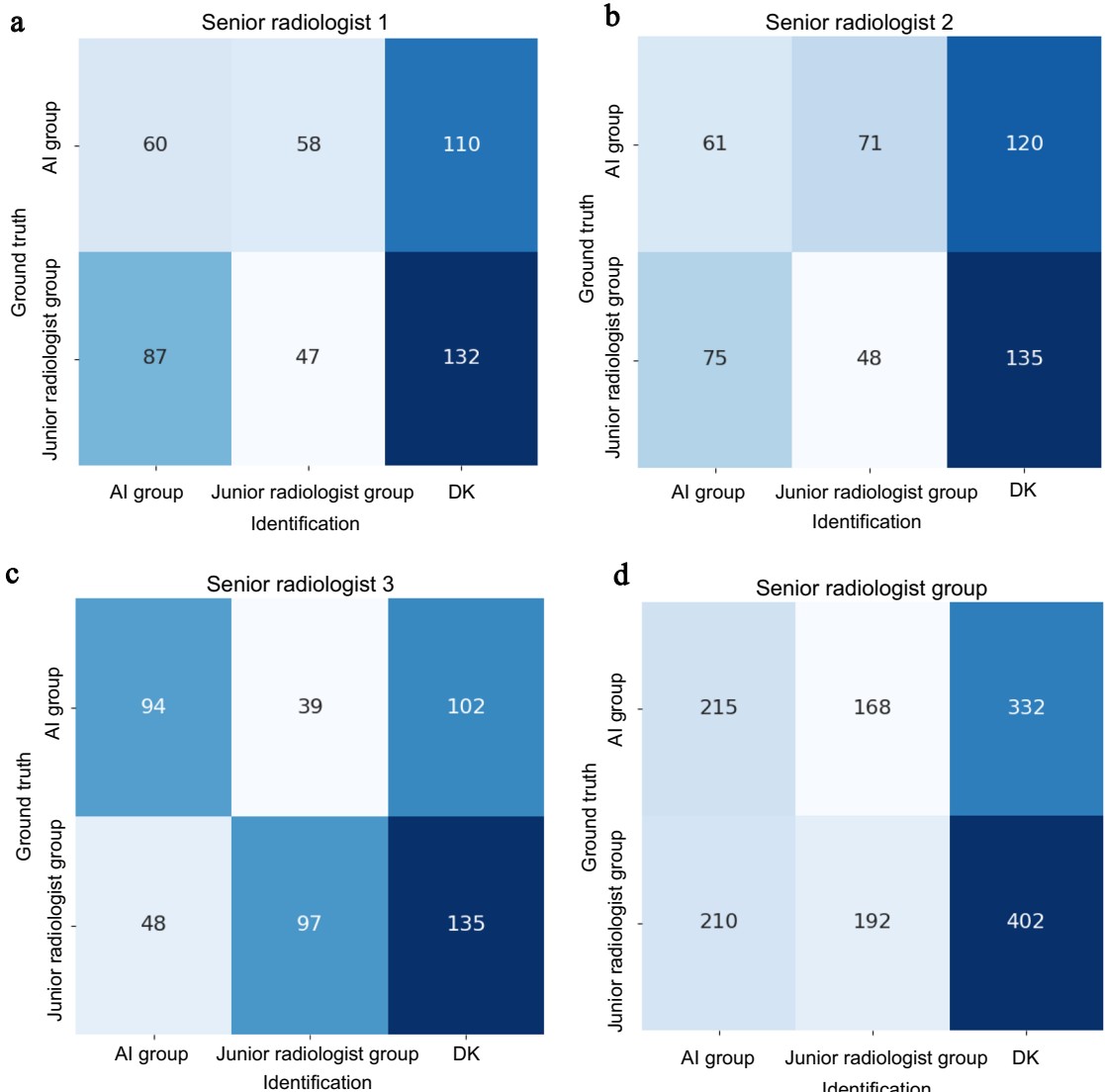

**Fig. 6 | The metric of source identification. a–c** The source identification by three senior radiologists, while **d** the combined source identification of the senior radiologist group. "Identification" refers to the radiologists' determination of whether the source is from AI or junior doctors, while "Ground truth" indicates the actual source of the data. "DK" denotes cases where the source could not be determined. Source data are provided as a Source data file.

radiologists were first tasked with determining whether the materials were AI-generated and then performed secondary diagnoses based on the provided information (Fig. 1e). Figure 6 illustrates the evaluation of the identification of the source of diagnostic materials. The senior radiologists correctly identified the source in 407 cases (26.8%), made incorrect identifications in 378 cases (24.9%), and were uncertain in 734 cases (48.3%). The degree of blinding in the trial was assessed using Bang's Blinding Index[26], a metric where 0 represents perfect blinding and values of −1 or 1 indicate complete unblinding. Typically, a blinding index between −0.2 and 0.2 is considered to reflect good blinding. In this study, the blinding index was 0.007 for the AI group and −0.031 for the junior radiologist group. Additionally, the bootstrapped confidence interval remained consistently within the range of −0.2 to 0.2, with no statistical significance ($P$ value > 0.49), further supporting the adequacy of blinding in the trial.

Table 3 shows the outcome of secondary diagnoses. The senior radiologists revised the initial diagnoses for 42 cases (5.5%) in the AI group compared with 158 cases (20.7%) in the junior radiologist group (difference −15.2%, 95% CI: −18.5% to −11.9%, $P$ value < 0.001 for superiority). This indicated that under blinded and randomized conditions, the AI group demonstrated better diagnostic accuracy. Moreover, the secondary diagnoses aligned with the original gold standard in 738 cases (97.5%) in the AI group compared with 746 cases (97.9%) in the junior radiologist group (difference −0.4%, 95% CI: −2.1% to 1.2%, $P$ value = 0.595 for superiority). The three senior radiologists took an average of 6.00 s (IQR: 5.20–7.05) to make secondary diagnoses based on the materials provided by the AI, and 8.00 s (IQR: 6.45–11.55) for those based on the materials from the junior radiologist group. The mean difference in time between the two groups was −2.79 s (95% CI: −3.87 to −1.70 s, $P$ value < 0.001). The shorter time required for diagnoses based on the materials provided by the AI group can be attributed to the fact that NCLS is able to provide more precise diagnostic materials.

### Evaluation of an AI system on blind sweeping data
To simulate a clinical environment with a shortage of experienced radiologists, we further validated our system using blind sweeping data. We prospectively collected 111 cases of blind sweeping data, where operators performed blind scans using a pre-set US probe intensity and were not allowed to view the US screen to adjust their

**Table 3 | Outcome of secondary diagnoses in a blind, randomized trial**

| Outcome | AI group (N = 757) | Junior radiologist group (N = 762) | Mean difference (95% CI) | P value |
|---|---|---|---|---|
| Initial result change | 42 (5.5%) | 158 (20.7%) | −15.2% (−18.5% to −11.9%) | <0.001 |
| Gold standard Consistency | 738 (97.5%) | 746 (97.9%) | −0.4% (−2.1% to 1.2%) | 0.595 |
| Secondary diagnosis time(s), median (IQR) | 6.00 (5.20–7.05) | 8.00 (6.45–11.55) | −2.79 (−3.87 to −1.70) | <0.001 |

Initial result change refers to the discrepancy between the secondary diagnosis by senior radiologists and the initial diagnosis made by AI or the junior radiologist group. Gold standard consistency refers to the agreement between the secondary diagnosis by senior radiologists and the gold standard diagnosis for each case. Secondary diagnostic time refers to the time spent by senior radiologists to make a diagnosis based on the provided materials. The superiority of initial result change and gold standard consistency was assessed using two-sided Pearson's chi-square tests (df = 1). Secondary diagnostic time was compared using a one-sided Welch's $t$-test under the directional hypothesis that AI assistance reduces diagnostic time. Source data are provided as a Source data file.

sweeping technique during the process (Fig. 1f). All blind sweeping CUS videos were independently reviewed by two senior radiologists and their diagnoses results were used as the gold standard, which includes 7 severe cases and 4 cases that were excluded due to being unsuitable for diagnosis. Figure 7a, b shows examples of the extracted standard views. Blind sweeping CUS videos were input to NCLS and extracted multiple standard views. Specifically, 84 cases had six standard views extracted, 24 cases had five standard views, 2 cases had four standard views, and 1 case had three standard views, which was also among the 4 cases excluded from diagnosis. Two senior radiologists were also recruited to provide qualitative assessments of the extracted standard views. The scoring criteria are outlined in Supplementary Table 12, which includes factors such as view correctness, detection accuracy, anatomical recognition, and clinical value. For Radiologist 1, the average scores were 3.8, 4.1, 4.1, and 4.0, respectively, for these criteria. For Radiologist 2, the corresponding average scores were 3.2, 3.3, 3.5, and 3.4. Statistically significant differences were observed for each score ($P$ value < 0.001). Figure 7c shows the qualitative scores made by the senior radiologists, and these findings suggest that the standard views extracted by the system from the blind sweeping data generally meet the requirements for clinical diagnosis.

Figure 7d shows the ROC curve, and Fig. 7e presents the confusion matrix for the NCLS on the blind sweeping data. Our system predicted the cases deemed unsuitable for diagnosis in the gold standard as non-severe, and achieved a sensitivity of 1.000 (8 out of 8), specificity of 0.961 (99 out of 103), F1-score of 0.800 and an AUC of 0.991 in the remaining blind sweeping data, indicating its potential for clinical applications, particularly in low-resource settings.

## Discussion

In this study, we presented NCLS, a pioneering system for identifying severe neonatal cerebral lesions that may require further intervention among neonates at high risk of brain injury, based on dynamic CUS images. Key findings of our study are discovered as follows: Firstly, the NCLS can automatically extract standard views of neonatal brain images as well as detect severe cerebral lesions with high accuracy from CUS videos; Secondly, the system also helps junior radiologists improve their diagnostic performance while reducing time consumption; Thirdly, the senior radiologists are not able to differentiate the standard views of CUS images obtained by the NCLS from those obtained by sonographers; Last but not least, the NCLS can potentially determine the standard views of CUS images and make correct diagnosis via blind sweeping US videos. All these findings are very important because they demonstrate that this tailored system can potentially streamline the screening process of detecting severe cerebral lesions among neonates at high risk of brain injury, as well as reduce the workload of radiologists. Thus, with the help of the NCLS, the healthcare of neonates at high risk of brain injury might be improved, especially in low- and middle-income countries or regions with limited medical resources.

Previous studies have confirmed the effectiveness of using DL models for automated diagnosis of CUS images. Dongwoon et al.[27] employed simple models like AlexNet and VGG16 for binary classification (normal vs. abnormal) of CUS coronal and sagittal images. Kim et al.[28] utilized DL algorithms to perform a binary classification task for IVH detection based on MSV images. Tahani et al.[22] used DL models for binary classification of normal and abnormal CUS images across multiple coronal views of extremely premature infants at different developmental stages. However, these studies have some limitations: first, they focused on classifying individual images as normal or abnormal, yet many mild abnormalities can resolve on their own[24]. Therefore, the focus should be on identifying severe abnormalities, such as high-grade IVH. Additionally, a single image is limited in providing a comprehensive diagnosis of neonatal cerebral lesions. Second, the performance of these DL models relies on high-quality CUS images and lacks prospective validation. In contrast, we propose the NCLS, which uses a multi-view combined diagnostic strategy, combining four to six standard coronal and sagittal views from the same neonate for a comprehensive diagnosis. Moreover, we introduce a classification scheme that further distinguishes between severe and non-severe cases based on the presence or absence of brain injury and its severity[5]. The NCLS has been validated with prospective clinical data and blind-sweep data as well, proving its reliability and effectiveness.

Several advantages of the NCLS should be emphasized. First, the NCLS showed robust diagnostic performance in detecting severe cerebral lesions across diverse test sets. This might be due to the fact that we did not restrict the scanners and probes that were used to acquire the developed set. We did not restrict specific sonographers who performed the CUS either, which might also increase the generality of our system. Notably, the NCLS identified all grade III and IV IVH in both internal and external test sets. In contrast, both junior and mid-level radiologists had several cases of misdiagnosis or missed diagnoses in grade II and grade III IVH cases, suggesting a lack of clear diagnostic standards among them. As a screening tool, this is very important for the NCLS not to miss any severe IVH, which is dramatically associated with long-term neurodevelopment disorders[7,29]. In addition, the performance of the NCLS in detecting severe cerebral lesions is superior to that of all junior radiologists in the external set. Furthermore, with the assistance of NCLS, the diagnostic sensitivities of all nine junior radiologists were greatly elevated in both test sets. This is very critical because a higher sensitivity might benefit both neonates suspected with severe cerebral lesions and their parents for timely intervention and early preparation for the potential negative outcomes. All these findings underscore the potential usefulness of the NCLS in screening severe cerebral lesions among neonates at high risk of brain injury in real clinical settings.

Second, the NCLS is interpretable because it not only provides standard views of CUS images with the diagnosis but also generates heatmaps for all standard views. The NCLS showed a powerful capacity to automatically extract standard views from CUS videos, even those obtained through blind sweeping. These standard views contain sufficient information for radiologists to make accurate diagnoses. Furthermore, the heatmaps consistently highlighted the anterior horn and body of the lateral ventricles across various disease types, aligning with critical regions for clinical diagnosis. Also, the heatmaps show good alignment with the areas marked by the radiologist, particularly in the

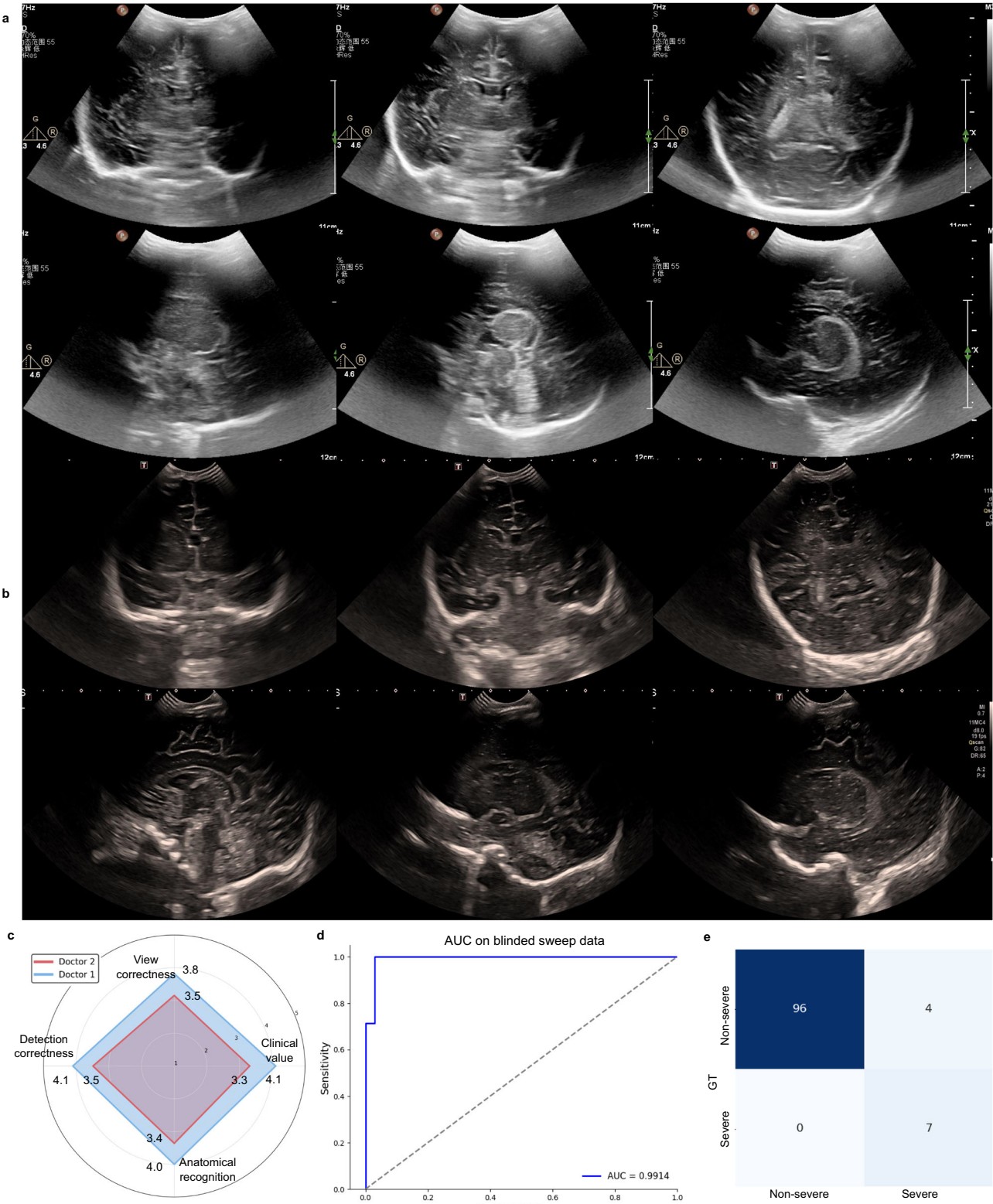

**Fig. 7 | Performance of the NCLS on blind sweeping data. a, b** The extracted standard views of NCLS on blind sweeping CUS videos. **c** Qualitative assessment of the extracted standard views by two senior radiologists, including view correctness, detection correctness, anatomical recognition, and clinical value. **d** ROC curve of NCSL. **e** Confusion matrix. Source data are provided as a Source data file.

MSV and coronal BV. With this capacity, even for those cases that the NCLS missed, the radiologists could make the right diagnosis by reviewing the standard views provided by the NCLS. Therefore, just as our results showed, the NCLS can not only help the junior radiologists

improve the diagnostic accuracy but also increase the agreement of them (Kappa values from 0.3005 and 0.3950 to 0.7034 and 0.7526).

Our study also showed that the standard views provided by the NCLS are quite similar to those provided by junior radiologists, and the

senior radiologists cannot distinguish between them. Thus, the NCLS might replace the junior radiologists or sonographers to provide standard views from CUS videos for further review with much less time consumption. Given that lots of time and energy are required for the tedious manual processing of CUS videos for junior radiologists or sonographers in their routine work, the results achieved by our system are encouraging.

Our study also demonstrated that the NCLS is able to process blind sweeping videos effectively, even under challenging conditions like blurriness, shaking, anatomical distortion, and fast probe movement. Moreover, the system was capable of providing accurate diagnoses based on these standard views. As we know, there is a lack of sonographers in low- and middle-income countries where medical resource is limited, such as those located in Africa or the Middle East[30]. It is said that there is a severe shortage of sonographers even in the United States[16]. The NCLS provides a potential pathway to solve this problem in such conditions. With the application of NCLS, we can employ novices to obtain CUS videos by blind sweeping, which are then given to the NCLS for further automatic analysis. A similar study has confirmed the usefulness of the DL algorithm in solving gestational age estimation via blind US sweeps in low-resource settings[31,32].

In this study, we used the Papile grading system rather than the more recent Volpe grading system, primarily considering the widespread applicability of the Papile grading system in China and the consistency of multi-center validation standards. However, we also performed validation on the Volpe grading system data to assess the robustness of NCLS. The outcomes showed that the prognosis under both grading systems is consistent with clinical prognosis.

Several limitations exist in this study. First, the NCLS still missed some severe cases, especially PVL. To address this issue, we plan to expand data collection and explore additional methods to improve screening effectiveness from a multimodal perspective, incorporating demographic metadata and Doppler US images. Existing work has also demonstrated that metadata used in traditional machine learning can yield good results in predicting the severity of IVH and mortality rates[33,34]. Second, the system is currently limited to binary classification, distinguishing only between intervention and non-intervention cases. It is not yet capable of diagnosing more specific conditions. Future exploratory plans will focus on identifying the specific diseases of newborns diagnosed for intervention. Additionally, since the training set was exclusively sourced from SZCH, we plan to collect a more diverse and generalized set from multiple centers in the next phase. Moreover, due to resource constraints, pathological gold standards and MRI confirmations were not consistently accessible. Therefore, consensus interpretation of pediatric radiologists was used as a practical, albeit less ideal, ground truth. In the future, we will continue to iterate and improve this CUS diagnostic system to enhance its performance, with plans to embed it into portable US devices for testing. Last but not least, this study did not reveal detailed information about the individuals who acquired the CUS images. Variations in operator experience could impact the generalizability of the NCLS.

In conclusion, we developed a robust deep learning system (NCLS) that can automatically extract standard views and make diagnoses from CUS videos. We demonstrated that this system could assist junior radiologists to elevate their diagnostic performance in screening severe cerebral lesions among neonates at high risk of brain injury. This system also showed potential capacity to replace junior radiologists to provide standard views from even blind sweeping CUS videos. Our proposed system might be useful to deploy in multiple application scenarios, such as intelligently detecting critical cerebral lesions in hospitals where medical resources are limited, and replacing sonographers to extract standard views of CUS images in hospitals where there is a shortage of sonographers. Future work will focus on addressing the identified limitations and conducting well-designed prospective studies to evaluate the effectiveness of our proposed system in screening neonatal cerebral lesions in real clinical settings.

## Methods

### Data collection and preprocessing

This study was conducted as a retrospective and prospective analysis, approved by the institutional review boards of each participating institution. The retrospective development set and prospective internal test set were obtained from SZCH (committee number 202312702), while the prospective external test set was collected from Guangzhou Panyu District Maternal and Child Health Care Hospital (IIT2023-17-02), Sichuan Provincial Maternity and Child Health Care Hospital (20240205-013), and Changsha Hospital for Maternal and Child Health Care (EC-20240102-09). The prospective section was registered at the Chinese Clinical Trial Registry (Registration number: ChiCTR2400079819). Consent was obtained from the guardians of all participants.

All CUS data were acquired by sonographers, junior radiologists, or trainees with 1–3 years of experience in pediatric ultrasound and stored in Digital Imaging and Communications in Medicine format. We first converted the images and videos to PNG and MP4 formats, respectively. Next, we applied a maximum connected domain image processing technique to extract the ROI from the CUS data, removing any information bars and other unrelated identifiers. Prior to training and validation, all data were anonymized.

We have collected neonatal CUS data since 2021, covering conditions such as normal, IVH (Grades I–IV), ependymal cyst, ventricular dilation, PVL, and hydrocephalus. The exclusion criteria for the CUS data were as follows: (1) incomplete CUS examinations; (2) extremely low-quality or unusable CUS data; (3) unclear diagnostic results; and (4) data from other ultrasound modalities, such as Color Doppler images. We also recorded key information for each examination, including the case ID, examination date, sex, age at examination, GA, BW, Apgar score, and the ultrasound equipment and probe. The ultrasound equipment used for the CUS data acquisition included GE HealthCare, EDAN, Mindray, Wisonic, Voluson, Philips Medical Systems, Toshiba, Canon, and Siemens. All this information is summarized in Table 1, and Supplementary Tables 1 and 2.

### Development of NCLS

The development of the NCLS was carried out in two stages. Stage 1 focuses on building the view extraction module, which includes a detection model and a candidate frame scoring algorithm. Stage 2 involves developing the diagnostic module, which includes a model capable of processing multi-view data.

### Stage 1: Building the view extraction module

We established the view extraction module that employs rotated bounding box object detection to identify anatomical structures within each frame of CUS videos. Based on the detection results, we implemented the candidate frame scoring algorithm to facilitate the extraction of standard views.

**Development of the object detection model.** We used an advanced end-to-end DL model, known as the Real-Time Detection Transformer (RT-DETR)[35], as shown in Supplementary Fig. 4a. This model utilizes ResNet50[36] as its backbone and incorporates a hybrid encoder architecture that effectively separates intra-scale interactions from cross-scale features, thereby enabling rapid and precise real-time detection. To better detect anatomical structures that may appear rotated in the CUS images, we incorporated a rotational angle dimension into the bounding box representation. This representation consists of five

parameters: center coordinates, width, height, and rotation angle (which ranges from −90° to 90°). The output from the encoder underwent a top-k selection process, and the resulting memory was used as the initialized anchors and contents, which were then passed into the decoder. The decoder consisted of a matching part and a denoising part. The matching part ensured that the predicted detection boxes corresponded to the anatomical structures, and the denoising part enhanced the robustness of the model. We used Varifocal Loss for the classification loss to better align the classification and regression tasks, and L1 Loss along with the DIoU Loss for bounding box regression to improve localization precision.

Following the official RT-DETR recommended hyperparameter settings, all CUS images were resized to 640 × 640. The training strategy—including learning rates, learning rate schedule, and most data augmentations—adhered to these defaults to ensure training stability and optimal performance. To further enhance data diversity and model robustness, we incorporated Mosaic and Mixup augmentations. Training was performed using PyTorch 2.1 on four Nvidia A6000 GPUs, with a batch size of 16 (4 samples per GPU).

**Candidate frame scoring algorithm.** We also developed the candidate frame scoring algorithm specifically to extract standard views from CUS videos based on the detected anatomical structures. Supplementary Fig. 4b shows the process of the candidate frame scoring algorithm, and Supplementary Table 13 shows the candidate frame selection criteria and base score of each structure. Specifically, each frame could either be sent to the candidate queue for the relevant standard view or left without further processing. Subsequently, we calculated the score of each frame in the queue based on all detected anatomical structures of this frame, and formulate as follows:

$$\text{Score} = \sum \text{Anatomical base score} \times \text{confidence}$$

Moreover, the Density-Based Spatial Clustering of Applications with Noise[37] algorithm was applied to ascertain the scanning direction of the CUS videos, and to eliminate candidate frames that did not satisfy the established criteria, as shown in Supplementary Fig. 5. We clustered candidate frames based on the Euclidean distance between score and frame index and identified the scanning direction of the video based on clusters in different standard planes. After filtering the frames, we sorted the frames in each candidate frame queue by score and selected the frame with the highest score as the standard view frame for the CUS video. Supplementary Fig. 6 displays the detection results and the candidate frame scoring process.

**Stage 2: Developing the diagnostic module**
We further developed the multi-view combined diagnostic module to screen for cerebral lesions.

**Diagnostic model development.** The diagnostic model framework is shown in Supplementary Fig. 7. We used ConvNext[38] as the feature extraction backbone. Multiple standard views from the same neonate were combined and fed into the backbone. The extracted features were, on the one hand, input into the image diagnostic classification head to perform multi-label classification for single-image prediction; on the other hand, these features were padded and concatenated with a class token and fed into a multi-head self-attention block for feature fusion. The first position of the fused output was then fed into the case diagnostic classification head to obtain the diagnostic result of whether cases were severe or not.

We implemented an ensemble diagnostic strategy to integrate outcomes from both the multi-label and binary classification heads. The multi-label head identified a neonate as severe if at least one image exhibited PVL, hydrocephalus, or a combination of IVH and ventriculomegaly. Conversely, the binary head provided a direct case-level prediction of severity. When the binary head predicted non-severe but the multi-label head suggested severe findings, we applied an additional rule-based refinement. Specifically, we counted the number of abnormal images in the case—defined as any image not predicted to be exclusively normal. If the multi-label predictions included severe findings and the number of abnormal images was two or more, the case was ultimately classified as severe.

For the multi-label classification of image predictions, we used binary cross-entropy loss. For the binary classification task of patient diagnosis, we adopted a class-balanced cross-entropy loss[39] to address the extreme data imbalance by re-weighting. Additionally, we utilized a regularization called RegMixup[40] to mitigate data uncertainty in the training period. We also implemented Test-Time Augmentation to enhance prediction robustness by averaging results from multiple augmented versions of the input images in the inference period.

We performed a grid search on the first fold of the validation set to identify the optimal combination of hyperparameters, including image size, number of epochs, learning rate, and batch size. The top 10 configurations are presented in Supplementary Table 14. Based on the best-performing setup, all CUS images were resized to 256 × 256, and various augmentations were applied during training, including flipping, grayscale conversion, rotation, and Gaussian blur. The model was trained for 45 epochs with a batch size of 16 using the AdamW optimizer, combined with a warm-up strategy and cosine learning rate decay. Training was conducted on a single Nvidia A6000 GPU using PyTorch version 2.1.

**Model validation**
**Ablation study.** We conducted an ablation study on the development test set to evaluate the efficacy of our proposed methods. The baseline model is a ConvNext architecture with multi-view fusion followed by binary classification. As shown in Supplementary Table 15, the integration of the multi-label classification head resulted in a significant improvement in diagnostic accuracy. This gain is attributed to the enhanced ability of the model to discern fine-grained diagnostic information during training. In contrast, the diagnostic fusion strategy provided limited improvements. Nevertheless, this module was retained in the final model to mitigate the potential for missed diagnoses.

**SOTA comparison.** We compared several classic and advanced classification models and employed the hyperparameters as recommended in the original publications. As shown in Supplementary Table 15, our proposed model with incorporating ensemble diagnosis achieved superior performance, yielding the highest AUC of 0.9725, SEN of 0.9317, and NPV of 0.9885. These findings demonstrate the outstanding ability to distinguish between cases of severe brain injuries.

**Evaluation metrics**
For stage 1, the detection performance of NCLS was evaluated by assessing mAP, FPS, and the qualitative score assessment of extracted standard views, where mAP represents the average precision across different recall levels. For stage 2, the performance of NCLS and that of all radiologists were evaluated by assessing the sensitivity, specificity, PPV, NPV, F1-score, and AUC with two-sided 95% CIs. Formulas as follows:

$$\text{Sensitivity} = \frac{\text{TP}}{\text{TP} + \text{FN}}$$
$$\text{Specificity} = \frac{\text{TN}}{\text{FP} + \text{TN}}$$
$$\text{PPV} = \frac{\text{TP}}{\text{TP} + \text{FP}}$$
$$\text{NPV} = \frac{\text{TN}}{\text{TN} + \text{FN}}$$
$$\text{F1} - \text{score} = 2 \cdot \frac{\text{Precision} \times \text{Recall}}{\text{Precision} + \text{Recall}}$$

We will also record the time taken by the NCLS for the entire screening process and compare it with the time required by radiologists.

### Statistical analyses

In this study, categorical variables were expressed as counts and percentages, and continuous variables were represented as median (95% CI), with the 95% CIs of all continuous variables calculated using the Clopper-Pearson method. For comparisons between different groups in the development dataset, the chi-square test was used for the significance testing of categorical variables, and the Mann–Whitney $U$ test was used for continuous variables. For comparisons of AUC performance between AI and radiologists, as well as between junior radiologists with and without AI assistance, a paired, one-sided Wilcoxon signed-rank test was used based on reader-level $\Delta$AUC values. In the blind and randomized trial, Bang's blinding index was used to assess the degree of blinding. The superiority of the correction rate and gold standard consistency was tested using two-sided Pearson chi-square tests, and secondary diagnostic time was compared using a one-sided Welch $t$-test under the directional hypothesis that AI assistance reduces diagnostic time. For subgroup analyses, differences in AUC between probe types were tested using the independent-sample DeLong test (two-sided), and pairwise comparisons across GA groups were conducted using the independent-sample DeLong test with Benjamini–Hochberg correction for multiple comparisons. Additionally, in the assessment under Papile and Volpe grading systems, AUC differences across GA groups were evaluated using two-sided independent-sample DeLong tests, also corrected using the Benjamini–Hochberg method. Agreement between AI and different radiologists was assessed using Cohen's kappa, and inter-rater consistency within the radiologist group was evaluated using Fleiss' kappa. In the qualitative scoring assessment, a paired $t$-test was used to evaluate the significance of differences between two radiologists across four scoring items.

### Reporting summary

Further information on research design is available in the Nature Portfolio Reporting Summary linked to this article.

## Data availability

The ultrasound images and videos used in this study are not publicly available due to hospital regulations and ethical restrictions protecting patient privacy. Qualified researchers may request access to de-identified data for non-commercial academic use. Requests should be submitted to the corresponding author and will require a data use agreement approved by the institutional ethics board of Shenzhen Children's Hospital. Access will only be granted to researchers affiliated with recognized institutions, with documented expertise in medical imaging or related fields, and with a clear scientific purpose aligned with the original study. The minimum dataset required to interpret and replicate the key findings is available under these access conditions. Access requests will be reviewed within 2 months of submission. Source data are provided with this paper.

## Code availability

The codes are available for scientific research and non-commercial use on GitHub at https://github.com/Je1zzz/Neonatal_cerebral_lesions_screening_NCLS. A citable version with a DOI is available via Zenodo at https://doi.org/10.5281/zenodo.15772754[41].

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

## Acknowledgements

This study was supported by Guangdong High-level Hospital Construction Fund at Shenzhen Children's Hospital high-level hospital medical platform project (Shenzhen Children's Hospital Medical Science Education [2023] No. 7; awarded to L.Z.), National Natural Science Foundation of China (No. 82271996; awarded by L.Z.), National Natural Science Foundation of China (No. 12326619; awarded to D.N.), Frontier Technology Development Program of Jiangsu Province (No. BF2024078; awarded to D.N.), Natural Science Foundation of Hunan province (No. 2022JJ30318; awarded to X.D.) and Scientific Research Project of Hunan Provincial Health Commission (No. W20243151; awarded to X.D.). We sincerely appreciate all members of the Department of Ultrasound, and Neonatology, Shenzhen Children's Hospital; Department of Ultrasonography, Changsha Hospital for Maternal and Child Health Care; Sichuan Provincial Woman's and Children's Hospital/The Affiliated Women's and Children's Hospital of Chengdu Medical College; Panyu Maternal and Child Care Service Centre of Guangzhou; Ultrasound Department of Longhua District Maternal and Child Healthcare Hospital; Department of ultrasound, Shenzhen Baoan Women's and Children's Hospital; Department of Medical Ultrasonics, Fujian Maternity and Child Health Hospital; and School of Biomedical Engineering, Shenzhen University.

## Author contributions

L.Z., D.N., and L.L. conceived the idea and designed the experiments. Z.L., H.Z., J.W., R.H., and L.Z. wrote and revised the paper. Z.L., H.Z., J.Z., F.X., and Z.S. collected the data. H.Z. and J.W. developed the deep learning model and validation software. Z.L., H.Z., J.W., R.H., and L.Z. analyzed the data and experimental results. X.D. and Y.B. contributed to the methodology and were responsible for guiding the multi-center data collection. Q.L., Y.C., H.Y., and Z.W. collected the multi-center data.

## Competing interests

The authors declare no competing interests.
