## [Transparent Peer Review file · Nature Communications]

Deep Learning approach for screening neonatal cerebral lesions on ultrasound in China

Corresponding Author: Dr Luyao Zhou

Version 0:

Reviewer comments:

Reviewer #1

(Remarks to the Author)

In this work, the authors present a AI based system to help diagnostic of severe neonatal cerebral lesions on cranial ultrasound (CUS).

The first part of the system is a detection neural network that select three coronal and three sagittal views of the in the video that includes the main cerebral structures. These views are then given to a second neural network that label each view with a diagnostic label (normal, ventricular dilatation,...) and classify the potential anomaly as severe or non severe. These two intermediate diagnostics as then fused into a final diagnostic. The system has been trained on the video of ~1500 patient from the Schenzen hospital and tested on an internal dataset and a external dataset (from three different Chinese hospital). The system has been compared to a traditional diagnostic made by radiologist ranked with their degree of expertise. The capacity of the system to improve the diagnostic of junior radiologist has also been evaluated and the capacity of the system to replace junior radiologist in a two step diagnostic process (junior and then senior radiologist). Finally the system was evaluated within a blind sweeping context.

papers on the related problem of fetal brain imaging on US are missing in the state of the art.

For example:

- Computer-aided diagnosis for fetal brain ultrasound images using deep convolutional neural networks. Baihong Xie etal, Int J Comput Assist Radiol Surg, 2020.
- Standard fetal ultrasound plane classification based on stacked ensemble of deep learning models, Bala Krishna etal, Expert Systems with Applications 2024

quantitative SOTA comparison

The article lack some quantitative comparison to state of the art or at least an ablation study to show that the modification made by the authors over reference network are useful. For example, for the diagnostic part, does the head (multilabel classification and the binary classification and then the multilabel fusion) added to the convnext backbone add something over a classical convnext.

interpretability

The authors claims that their system "is interpretable because it can provide standard views of CUS images with the diagnosis".

To me, this is a very low interpretability criterion. The system could easily be made more interpretable if attribution/attention map as given for example by grad cam or integrated gradient would be given as an output of the system. It would also be interesting to include a study of these maps in the manuscript.

A natural question would be for example to see whether something is visible around the ventricles on the attribution maps when the view is classified as "ventricular dilatation". This would enable to detect potential shortcuts in the network caused by biases in the dataset.

In the description of their system, the authors present the hyperparameter they choose to use: input image size, data augmentation, number of epochs, initial learning rate, ...

How were these parameters chosen ? Was there some sort of tuning of optimization either manual or a grid search to adjust these parameters.

If yes, which dataset was used for these hyperparameters optimization.

The multi task fusion decision from binary classif (severe/non severe) and multilabel imagewize classification is not described.

Please add a description of this part.

Some figures (fig 1, 2 and 3) are too small. The reader needs to zoom to be able to read in the pdf version and is unreadable to me on a printed version.

(Remarks on code availability)

Reviewer #2

(Remarks to the Author)

This study presents an interesting application of deep learning techniques in the evaluation of neonatal brain lesions using ultrasound. The study methodology and the number of studies used to train the system appear to be more than satisfactory. I have a series of general observations on the use of this technique in ultrasound, in particular:

a. The main limitation of the study seems to be the binary classification of lesions as severe and non-severe. Grade III and IV intracranial hemorrhages (ICH) according to Papile, periventricular leukomalacia (PVL), and cases of hydrocephalus are considered severe. However, we know that there are extensive discussions regarding Papile's classification, and today different classifications, such as those proposed by J.J. Volpe or the consensus of the European group, are preferred. These classifications separately consider the presence of germinal matrix hemorrhage, the presence of blood in the ventricles (intraventricular hemorrhage, IVH), and the presence of associated parenchymal lesions, defined as periventricular hemorrhagic infarctions (PHI). It is possible that a Papile grade I hemorrhage is actually associated with a PHI, thereby making the overall condition severe, even though it would otherwise be considered prognostically favorable.

b. Additionally, according to Papile, the distinction between grade II and III hemorrhages is based on whether the intraventricular blood covers more or less than 50% of the ventricular lumen. As is well known to anyone working with ultrasound, this assessment is often arbitrary, difficult to perform, and likely of limited prognostic value. What truly matters is the extent of hydrocephalus resulting from IVH

Based on this last consideration, it may be obvious that an automated system could outperform the human eye in assessing whether the presence of blood in the ventricles exceeds 50%, but it remains to be seen whether this has real prognostic utility. The main issues with hemorrhagic lesions are intraparenchymal lesions (PHI) and hydrocephalus. While hydrocephalus can be easily evaluated using deep learning techniques, I wonder if the same applies to PHI when using Papile's classification system.

Furthermore, automatic recognition performs worst in assessing PVL, which could also be a significant issue, as severe PVL is prognostically relevant regardless.

Given the good results obtained in the binary evaluation using Papile's classification, I believe the system should be tested with a more updated classification that can more precisely define the severity level while still maintaining a simple binary system.

(Remarks on code availability)

Honestly, I was not able to install the application, and I don't think I would be able to replicate what the authors have done, as a clinical pediatric neuroradiologist, I do not have the expertise to use these applications.

I was able to watch demos on how the ultrasounds were performed. They appear to have been conducted correctly, although with a weak signal in the more superficial regions—an aspect that could at least partially explain the lower sensitivity in detecting PVL

Reviewer #3

(Remarks to the Author)

The study introduces a valuable deep learning-based system, Neonatal Cerebral Lesions Screening, designed to enhance the detection of severe neonatal cerebral lesions using cranial ultrasound. This development is particularly significant given the global shortage of experienced pediatric radiologists. While the study presents notable strengths, there are limitations and missing details that need to be addressed:

1. One of the major limitations of this AI tool is its reliance on subjective diagnostic information from radiologists based on pre-existing ultrasound scans. The lack of an objective gold standard, such as MRI validation, weakens the study's ability to confirm diagnostic accuracy. MRI could have served as a more definitive reference, especially in cases where mild abnormalities could progress into severe lesions.

2. Ultrasound imaging is highly operator-dependent, and image quality varies significantly based on the performer, machine settings, and scanning conditions. Despite this, the study provides limited details on how ultrasound scans were conducted. For example, the probe is only classified as linear vs. curved, without specifying frequency settings or additional technical parameters that influence image resolution.

3. Even in the prospective dataset collection (e.g., blind sweeping method), critical details are missing: Who performed the

ultrasound? What equipment was used? (Machine models, manufacturers, and software versions.) Which probe settings were applied? (Frequency, gain, depth settings.)

4. The study categorizes findings into "severe" and "mild" lesions, but this binary classification oversimplifies the clinical significance of intermediate cases. Grade I and II IVH are classified as mild, but these cases carry a risk of progression to higher-grade IVH, particularly in preterm infants. A three-tier classification—"normal," "mild/moderate," and "severe"—would be more clinically useful and practical.

5. Including clinical context in the AI system would have been helpful. For example, incorporating the patient's gestational age could significantly impact the interpretation of ultrasound findings.

Other issues:

1. The Introduction states that the dataset consists of 8,757 neonatal CUS images, yet the Results section presents different case numbers for internal and external validation. Additionally, the Introduction implies this is a single-institution study, while the Results indicate data collection from multiple hospitals. These inconsistencies should be clarified.

2. Some sentences need revision for grammatical accuracy. For example: "Since mild abnormalities..." or "Blind sweeping CUS videos were inputted to..."

3. Some figure descriptions are misleading or incorrect. For example: Supplementary Figure 1 is described as including a pie chart and violin plot, but these are not present in the figure.

(Remarks on code availability)

Version 1:

Reviewer comments:

Reviewer #1

(Remarks to the Author)

The manuscript is much better in its revised form.

All my comments and remarks have been addressed.

I recommend the acceptance of the paper.

(Remarks on code availability)

Reviewer #2

(Remarks to the Author)

The authors have responded more than satisfactorily to my comments, including in their analysis not only the old Papile classification but also the more modern Volpe classification. I have no further questions or comments.

(Remarks on code availability)

I have not reviewed the code

Reviewer #3

(Remarks to the Author)

1. Thank you for including technical details regarding the ultrasound acquisition protocols. However, a critical component that appears to be missing is information about the individuals who performed the ultrasound examinations. Specifically, it would be helpful to clarify whether sonographers, radiologists, or trainees performed the scans, and what level of training or experience they had prior to image acquisition. If these data are unavailable, I recommend addressing this limitation in the Discussion section.

2. The manuscript suggests that the observed heterogeneity in ultrasound data supports the generalizability of the AI model. While this is a potentially valuable observation, it would strengthen the manuscript to elaborate on whether any measures were taken to ensure or enhance generalizability during model development.

(Remarks on code availability)

We have carefully studied all the comments from the reviewers and made revision accordingly. We not only performed extra test to meet the acquirement from the reviewers but also improved the whole manuscript according to the editorial requests. All the revisions in the main manuscript were highlighted with underlines.

As follows are the point-by-point responses to the reviewers' comments.

Reviewer 1

In this work, the authors present an AI based system to help diagnostic of severe neonatal cerebral lesions on cranial ultrasound (CUS).

The first part of the system is a detection neural network that select three coronal and three sagittal views of the in the video that includes the main cerebral structures. These views are then given to a second neural network that label each view with a diagnostic label (normal, ventricular dilatation...) and classify the potential anomaly as severe or non-severe. These two intermediate diagnostics as then fused into a final diagnostic. The system has been trained on the video of ~1500 patient from the Shenzhen hospital and tested on an internal dataset and an external dataset (from three different Chinese hospital). The system has been compared to a traditional diagnostic made by radiologist ranked with their degree of expertise. The capacity of the system to improve the diagnostic of junior radiologist has also been evaluated and the capacity of the system to replace junior radiologist in a two-step diagnostic process (junior and then senior radiologist). Finally, the system was evaluated within a blind sweeping context.

Response:

Thank you for your recognition and positive feedback on our work. The following are the detailed responses to the comments.

Reviewer Comment 1.1

Papers on the related problem of fetal brain imaging on US are missing in the state of the art. For example:

- Computer-aided diagnosis for fetal brain ultrasound images using deep convolutional neural networks. Baihong Xie etal, Int J Comput Assist Radiol Surg, 2020.
- Standard fetal ultrasound plane classification based on stacked ensemble of deep learning models, Bala Krishna etal, Expert Systems with Applications 2024

Response:

We thank the reviewer for this comment. We have reviewed the literatures which you recommended and reflected on the connection between fetal imaging and our study.

We added on page 2 (see Introduction section, paragraph 3):

"Unlike human radiologists who rely on subjective experience, DL models analyze images by identifying deep and complex patterns, which allows for faster, more consistent and objective diagnoses. This capability could help alleviate the shortage of trained radiologists. In recent years, the development of DL for fetal cranial imaging^{1,2} has refined prenatal diagnosis, laying the foundation for early intervention in neonatal brain disorders. For instance, a recent study has focused on binary classification of CUS images (e.g., normal vs. abnormal) in very preterm infants³, but the clinical value of such models is limited. "

Reviewer Comment 1.2

Quantitative SOTA comparison

The article lacks some quantitative comparison to state of the art or at least an ablation study to show that the modification made by the authors over reference network are useful. For example, for the diagnostic part, does the head (multilabel classification and the binary classification and then the multilabel fusion) added to the convnext backbone add something over a classical convnext.

Response:

We fully agree with reviewer that the addition of more quantitative comparison experiments could further validate the proposed framework. Subsequently, we conducted state-of-the-art (SOTA) comparison and ablation experiments to evaluate the effectiveness of the proposed module. Multiple SOTA models were selected and trained using the hyperparameter configurations reported in their respective original publications, with evaluation performed on the development test set. Results are reported in the Supplementary Materials (See Supplementary Table 14):

Supplementary Table 1 | Comparison table of SOTA models.

Method	SEN	SPE	PPV	NPV	F1-score	AUC
ResNet-50 ⁴	0.7941	0.9712	0.7714	0.9747	0.7826	0.9623
ViT-Base ⁵	0.8529	0.9388	0.6304	0.9812	0.7250	0.9593
MambaOut ⁶	0.7941	0.9712	0.7714	0.9747	0.7826	0.9586
SigLipV2 ⁷	0.8235	0.8489	0.4000	0.9752	0.5385	0.9034
MedViT ⁸	0.8824	0.8849	0.4839	0.9840	0.6250	0.9576
MedMamba ⁹	0.7941	0.8993	0.4909	0.9728	0.6067	0.9544
Query2label ¹⁰	0.8235	0.9424	0.6364	0.9776	0.7179	0.9471
Baseline	0.8235	0.9604	0.7179	0.9780	0.7671	0.9709
+ Multi-label head	0.9098	0.9037	0.6810	0.9780	0.7789	0.9725

+ Ensemble diagnosis(**Ours**) **0.9118** 0.9317 0.6200 **0.9885** 0.7381 **0.9725**

This table compares the performance of SOTA models, including ResNet-50 and ViT-Base (classic SOTA models), MambaOut and SigLipV2 (SOTA classification models), MedViT and MedMamba (SOTA classification models in the medical field), and Query2Label (a SOTA multi-label classification model). The baseline model is based on the ConvNext, which is further enhanced with a multi-view feature fusion module. It is further improved with a multi-label head and ensemble diagnosis, where "+Ensemble diagnosis" represents our proposed method. Metrics include Sensitivity (SEN), Specificity (SPE), Positive Predictive Value (PPV), Negative Predictive Value (NPV), F1-score, and Area Under the Curve (AUC). The best result for each metric is highlighted in bold.

We added a subsection to discuss the ablation study and SOTA comparison (see page 16):

"**Ablation study.** We conducted an ablation study on the development test set to evaluate the efficacy of our proposed methods. The baseline model is a ConvNext architecture with multi-view fusion followed by binary classification. As shown in Supplementary Table 14, the integration of the multi-label classification head resulted in a significant improvement in diagnostic accuracy. This gain is attributed to the enhanced ability of the model to discern fine-grained diagnostic information during training. In contrast, the diagnostic fusion strategy provided limit improvements. Nevertheless, this module was retained in the final model to mitigate the potential for missed diagnoses.

SOTA comparison. We compared several classic and advanced classification models and employed the hyperparameters as recommended in the original publications. As shown in Supplementary Table 14, our proposed model with incorporating ensemble diagnosis achieved superior performance, yielding the highest AUC of 0.9725, SEN of 0.9317 and NPV of 0.9885. These findings demonstrate the outstanding ability to distinguish between cases of severe brain injuries."

We appreciate the reviewer's suggestion and we believe these quantitative comparison experiments and the ablation study clearly demonstrate the effectiveness of our proposed modifications over the baseline ConvNext model.

Reviewer Comment 1.3

The authors claims that their system "is interpretable because it can provide standard views of CUS images with the diagnosis".

To me, this is a very low interpretability criterion. The system could easily be made more interpretable if attribution/attention map as given for example by grad cam or integrated gradient would be given as an output of the system. It would also be interesting to include a study of these maps in the manuscript.

A natural question would be for example to see whether something is visible around the ventricles on the attribution maps when the view is classified as "ventricle dilatation". This would enable to detect potential shortcuts in the network caused by biases in the dataset.

Response:

We thank for the review's insightful comment. We applied Finer-CAM to the module before the multi-view feature fusion in the diagnostic model, and presented some representative cases, which included (a) normal case, (b) case with ventricle dilatation, (c) and (d) cases with hydrocephalus and IVH-III.

We added content to introduce and analyze the heatmaps (see page 6, Result section, Comparison of NCLS and radiologists part , paragraph 3):

“Extended Figure 2 shows the heatmaps generated from multiple views using Finer-CAM¹¹, and it reveals that the NCLS focuses on areas around the anterior horn and body of the lateral ventricles. Supplementary Figure 2 shows the regions of interest (ROIs) annotated by the radiologist and the heatmaps, demonstrating a high degree of overlap between the areas of focus identified by the NCLS and those identified by the radiologist.”

We corrected some unreasonable content and supplemented the description of heatmaps on page 11 (see Discussion section, paragraph 4):

“Second, the NCLS is interpretable because it not only provides standard views of CUS images with the diagnosis but also generates heatmaps for all standard views. The NCLS showed powerful capacity to automatically extract standard views from CUS videos, even those obtained through blind sweeping. These standard views contain sufficient information for radiologists to make accurate diagnoses. Furthermore, the heatmaps consistently highlighted the anterior horn and body of the lateral ventricles across various disease types, aligning with critical regions for clinical diagnosis. Also, the heatmaps show good alignment with the areas marked by the radiologist, particularly in the mid-sagittal view and coronal body view. With this capacity, even for those cases that the NCLS missed, the radiologists could make the right diagnosis by reviewing the standard views provided by the NCLS. Therefore, just as our results showed, the NCLS can not only help the junior radiologists improve the diagnostic accuracy but also increase the agreement of them (Kappa values from 0.3005 and 0.3950 to 0.7034 and 0.7526).”

We added heatmaps of the standard views (see Extended Figure 2):

Extended Figure 1. Heatmap for certain cases. **a** Normal cases. **b** Case with ventricle dilatation. **c and d** Cases with hydrocephalus and Grade-III IVH. The highlighted areas (red and yellow) represent the regions of significant attention by the model.

We added a figure of ROI annotated by the radiologist in supplementary material (see Supplementary Figure 2):

Supplementary Figure 1 | ROI annotated by the radiologist and the heatmaps. The first column shows the original images, the second column displays the ROIs of the ventricle (in red) and the choroid plexus or bleeding focus (in green). The third column presents the heatmaps.

We believe that the inclusion of heatmaps and ROI annotations significantly enhances the interpretability of our system, providing valuable insights into its decision-making process.

Reviewer Comment 1.4

In the description of their system, the authors present the hyperparameter they choose to use: input image size, data augmentation, number of epochs, initial learning rate, ... How were these parameters chosen? Was there some sort of tuning of optimization either manual or a grid search to adjust these parameters. If yes, which dataset was used for these hyperparameters optimization.

Response:

Thanks for your comments. We apologize for the lack of clarity in the hyperparameter selection. For the diagnostic model, we used the python library (Ray Tune) to perform hyperparameter search via grid search on one fold of the five-fold cross-validation set. The hyperparameters searched include learning rate (from 1×10^{-2} to 1×10^{-5}), batch size (1, 8, 16), image size (128, 256, 384) and the number of epochs (25, 45, 60). For data augmentation, we employed common augmentation strategies to simulate

various image variations and enhance data diversity, including rotation, grayscale adjustments, gaussian blur and flipping. During training, we also introduced RegMixup, which helped alleviate the long-tail issue to some extent. For the object detection model, to ensure training stability, we used the officially recommended hyperparameter settings. Additionally, we incorporated mixup and mosaic data augmentation to improve robustness.

We added a hyperparameter search result table, displaying the top 10 parameter combinations ranked by their scores (score = 0.7 * sensitivity + 0.3 * F1-score; see Supplementary Table 13):

Batch size	Epochs	Image size	Learning rate	Sensitivity	Specificity	F1-score
16	45	256	1×10^{-4}	0.9394	0.9793	0.9118
16	45	128	1×10^{-5}	0.9394	0.9534	0.8493
16	25	256	1×10^{-4}	0.9091	0.9793	0.8955
16	25	256	1×10^{-5}	0.9091	0.9793	0.8955
16	25	128	1×10^{-5}	0.9394	0.9378	0.8158
8	45	128	1×10^{-5}	0.9091	0.9689	0.8696
8	25	256	1×10^{-4}	0.8788	0.9845	0.8923
16	45	256	1×10^{-5}	0.8788	0.9845	0.8923
16	60	256	1×10^{-5}	0.8788	0.9793	0.8788
8	25	256	1×10^{-5}	0.8788	0.9793	0.8788

We revised the content to make the expression clearer (see page 15, Development of the object detection model section, paragraph 2):

“Following the official RT-DETR recommended hyperparameter settings, all CUS images were resized to 640×640. The training strategy—including learning rates, learning rate schedule, and most data augmentations—adhered to these defaults to ensure training stability and optimal performance. To further enhance data diversity and model robustness, we incorporated Mosaic and Mixup augmentations. Training was performed using PyTorch 2.1 on four Nvidia A6000 GPUs, with a batch size of 16 (4 samples per GPU).”

We provided an explanation of the hyperparameter selection and the results (see page 16, Diagnostic model development section, paragraph 4):

“We performed a grid search on the first fold of the validation set to identify the optimal combination of hyperparameters, including image size, number of epochs, learning rate, and batch size. The top 10 configurations are presented in Supplementary Table 13. Based on the best-performing setup, all CUS images were resized to 256×256, and various augmentations were applied during training,

including flipping, grayscale conversion, rotation, and Gaussian blur. The model was trained for 45 epochs with a batch size of 16 using the AdamW optimizer, combined with a warm-up strategy and cosine learning rate decay. Training was conducted on a single Nvidia A6000 GPU using PyTorch version 2.1.”

We hope this additional explanation clarifies the process of hyperparameter selection and optimization.

Reviewer Comment 1.5

The multi task fusion decision from binary classification (severe/non severe) and multilabel image-wise classification is not described. Please add a description of this part.

Response

Thank you. We apologize for the lack of clarity of the ensemble diagnostic strategy.

We added a detail description on page 15 (see Developing the diagnostic module section, paragraph 2):

“We implemented an ensemble diagnostic strategy to integrate outcomes from both the multi-label and binary classification heads. The multi-label head identified a neonate as severe if at least one image exhibited PVL, hydrocephalus, or a combination of IVH and ventriculomegaly. Conversely, the binary head provided a direct case-level prediction of severity. When the binary head predicted non-severe but the multi-label head suggested severe findings, we applied an additional rule-based refinement. Specifically, we counted the number of abnormal images in the case—defined as any image not predicted to be exclusively normal. If the multi-label predictions included severe findings and the number of abnormal images was two or more, the case was ultimately classified as severe.”

We believe this additional explanation clarifies the multi-task fusion decision process. Thank you for pointing out the need for further details, and we hope this revision enhances the understanding of our ensemble diagnostic strategy.

Reviewer Comment 1.6

Some figures (fig 1, 2 and 3) are too small. The reader needs to zoom to be able to read in the pdf version and is unreadable to me on a printed version.

Response

We thank the reviewer for their constructive comments on figure clarity. All figures

have been updated to comply with Nature Communications standards, including ensuring a resolution of 300 dpi for improved visual presentation.

Reviewer 2

This study presents an interesting application of deep learning techniques in the evaluation of neonatal brain lesions using ultrasound. The study methodology and the number of studies used to train the system appear to be more than satisfactory.

Response:

We thank the reviewer for acknowledging the contribution of our work. The following are the detailed responses to the comments.

Reviewer Comment 2.1

The main limitation of the study seems to be the binary classification of lesions as severe and non-severe. Grade III and IV intracranial hemorrhages (ICH) according to Papile, periventricular leukomalacia (PVL), and cases of hydrocephalus are considered severe. However, we know that there are extensive discussions regarding Papile's classification, and today different classifications, such as those proposed by J.J. Volpe or the consensus of the European group, are preferred. These classifications separately consider the presence of germinal matrix hemorrhage, the presence of blood in the ventricles (intraventricular hemorrhage, IVH), and the presence of associated parenchymal lesions, defined as periventricular hemorrhagic infarctions (PHI). It is possible that a Papile grade I hemorrhage is actually associated with a PHI, thereby making the overall condition severe, even though it would otherwise be considered prognostically favorable.

Additionally, according to Papile, the distinction between grade II and III hemorrhages is based on whether the intraventricular blood covers more or less than 50% of the ventricular lumen. As is well known to anyone working with ultrasound, this assessment is often arbitrary, difficult to perform, and likely of limited prognostic value. What truly matters is the extent of hydrocephalus resulting from IVH.

Based on this last consideration, it may be obvious that an automated system could outperform the human eye in assessing whether the presence of blood in the ventricles exceeds 50%, but it remains to be seen whether this has real prognostic utility. The main issues with hemorrhagic lesions are intraparenchymal lesions (PHI) and hydrocephalus. While hydrocephalus can be easily evaluated using deep learning techniques, I wonder if the same applies to PHI when using Papile's classification

system.

Given the good results obtained in the binary evaluation using Papile's classification, I believe the system should be tested with a more updated classification that can more precisely define the severity level while still maintaining a simple binary system.

Response:

Thank you for highlighting the potential limitation regarding to the Papile's classification. In this paper, we used the Papile classification because it is widely used across China for diagnosing neonatal intracranial conditions.

In response to your valuable suggestion, we have conducted additional validation of the Volpe's classification using the existing model and dataset. This included reclassifying the IVH data according to the Volpe grading method, reassessing them with the existing model, and analyzing neonatal prognosis statistics derived from MRI data.

We added a Sankey diagram to illustrate the IVH cases based on the Papile classification and showed the distribution changes of these cases after reclassification according to the Volpe grading method (see Supplementary Figure 3):

Supplementary Figure 2 | Reclassification of IVH Cases from Papile to Volpe Grading Systems across Datasets. **a** IVH grade redistribution in the development set. **b** IVH grade redistribution in the internal test set. **c** IVH grade redistribution in the external test set. All figures illustrate the distribution of IVH cases initially classified according to the Papile grading system (left) and their subsequent distribution after reclassification using the Volpe grading method (right) in the development set. Flow thickness represents the number of cases transitioning between grades.

From the figure we can know that in case of mild IVH (Grade I-II), there was a low consistency between the two grading systems. In contrast, in cases of severe IVH (Grade III-IV), there was a high level of consistency. Notably, the grading discrepancies in mild cases mainly stemmed from the interpretation of the degree of lateral ventricular enlargement and the extent of subependymal hemorrhage, indicating subjective judgment biases among radiologists when identifying subtle radiological features.

We added a statistical figure depicting the prognosis of IVH cases (see Supplementary Figure 4). The figures indicate that the prognostic trends under both grading systems are largely consistent and in line with clinical outcomes.

Supplementary Figure 3 | Prognostic Outcomes Under Two Grading Systems. a

Prognostic outcome under the Papile grading system. **b** Prognostic outcomes under the Volpe grading system. Each panel consists of three columns: the first column represents cases with a worsened condition, the second column represents cases with a recovered condition, and the third column represents cases with no follow-up.

We added the evaluation results of the model under both grading systems (see Supplementary Table 11). From the table, we can observe that the NCLS model demonstrates similar performance under both grading systems. The p-values are statistical measure of the difference in AUCs, calculated using the DeLong method. It is worth nothing that all p-values are greater than 0.1, which indicates that the differences in model performance between the two grading systems are not statistically significant, suggesting that the observed performance is likely due to random variability rather than the grading method itself.

Test set	Classification system	Classification				F1-score	AUC	P-value
		SEN	SPE	PPV	NPV			
Development test set	Papile	0.912	0.932	0.620	0.989	0.738	0.973	0.442
	Volpe	0.875	0.900	0.500	0.984	0.636	0.962	
Internal test set	Papile	0.875	0.924	0.500	0.988	0.636	0.981	0.748
	Volpe	0.833	0.939	0.577	0.983	0.682	0.981	

External	Papile	0.962	0.927	0.510	0.997	0.667	0.944	0.444
test set	Volpe	0.943	0.953	0.688	0.994	0.795	0.972	

We added on page 6 (see Result section, Comparison of NCLS and radiologists part, paragraph 3):

“Furthermore, we also conducted an analysis based on the Papile and Volpe grading systems for IVH. Supplementary Figure 3 shows the distribution changes of the original Papile graded IVH data after being reclassified according to the Volpe grading system. Supplementary Table 11 provides the evaluation results of the NCLS using both grading systems. Additionally, Supplementary Figure 4 illustrates the prognosis of IVH cases under both grading systems.”

We added a discussion of two grading system on page 12 (see Discussion section, paragraph 7):

“In this paper, we used the Papile grading system rather than the more recent Volpe grading system, primarily considering the widespread applicability of the Papile grading system in China and the consistency of multi-center validation standards. However, we also performed validation on the Volpe grading system data to assess the robustness of NCLS. The outcome showed that the prognosis under both grading systems is consistent with clinical prognosis.”

We hope that the additional validation and analysis help clarify the robustness of NCLS and address your concerns. We sincerely thank the reviewer for their valuable suggestion.

Reviewer Comment 2.2

Furthermore, automatic recognition performs worst in assessing PVL, which could also be a significant issue, as severe PVL is prognostically relevant regardless.

Response:

Thank you for pointing out the limitations of our NCLS system in assessing PVL. Ultrasound demonstrates considerable sensitivity in the detection of diffuse encephalomalacia and multifocal cystic PVL. As neonatal care improves, non-cystic PVL, a condition where ultrasound demonstrates limited sensitivity, is more common in preterm infants, thus reducing the incidence of cystic PVL (severe PVL). Consequently, the availability of training data has been reduced due to its declining incidence. Future work needs to collect more severe PVL cases from multiple centers to improve the performance in assessing PVL. New technologies, such as multimodal

diagnostics, will also be utilized, and we hope to overcome this challenge in the near future.

Reviewer Comment 2.3

Honestly, I was not able to install the application, and I don't think I would be able to replicate what the authors have done, as a clinical pediatric neuroradiologist, I do not have the expertise to use these applications.

I was able to watch demos on how the ultrasounds were performed. They appear to have been conducted correctly, although with a weak signal in the more superficial regions—an aspect that could at least partially explain the lower sensitivity in detecting PVL

Response:

We understand the challenges you faced with using the demo application. We have updated our demo code on the GitHub website to simplify the steps for potential users. For those unable to use the code, we have also provided a demonstration of the code execution process and results, to facilitate understanding of our system's operational workflow. Our GitHub website:

https://github.com/Je1zzz/Neonatal_cerebral_lesions_screening_NCLS

Reviewer 3

The study introduces a valuable deep learning-based system, Neonatal Cerebral Lesions Screening, designed to enhance the detection of severe neonatal cerebral lesions using cranial ultrasound. This development is particularly significant given the global shortage of experienced pediatric radiologists.

Response:

Thank you for your recognition and positive feedback on our work. The following are the detailed responses to the comments.

Reviewer Comment 3.1

One of the major limitations of this AI tool is its reliance on subjective diagnostic information from radiologists based on pre-existing ultrasound scans. The lack of an objective gold standard, such as MRI validation, weakens the study's ability to confirm diagnostic accuracy. MRI could have served as a more definitive reference,

especially in cases where mild abnormalities could progress into severe lesions.

Response:

Thanks for the comments. We fully acknowledge that MRI is the gold standard for diagnosing neonatal brain lesions. The suggestion to correlate our findings with MRI results to enhance diagnostic power is significant. However, in China, medical resources are relatively scarce, and the coverage of MRI equipment in tertiary hospitals is less than 60%^[1]. Furthermore, neonatal MRI requires sedation or anesthesia, which poses a considerable risk to preterm infants due to the potential for respiratory depression caused by sedative drugs. The high cost of MRI also places a substantial financial burden on average families. Therefore, clinical practice typically reserves MRI examinations for cases with significant progression observed through repeated ultrasound reviews, and this has resulted in a limited availability of MRI data. This challenge has also been highlighted in a published study on neonatal brain AI^[2]. We are currently collecting relevant MRI data for the next step of our research, at which point we will conduct a more comprehensive analysis of the availability of ultrasound diagnostics, the gold standard, and other related issues.

[1] National Bureau of Statistics of China. (2025). 2023 Statistical Monitoring Report on the China Children's Development Outline (2021–2030).

[2] Kim KY, Nowrangi R, McGehee A, Joshi N, Acharya PT. Assessment of germinal matrix hemorrhage on head ultrasound with deep learning algorithms. *Pediatr Radiol*. 2022 Mar;52(3):533-538. doi: 10.1007/s00247-021-05239-w. Epub 2022 Jan 22. PMID: 35064324.

We added an explanation regarding the selection of the gold standard for data on page 12 (see Discussion section, paragraph 8):

“Moreover, due to resource constraints, pathological gold standards and MRI confirmations were not consistently accessible. Therefore, a pediatric interpretation of radiologist was used as a practical, albeit less ideal, ground truth.”

We hope that our explanation clarifies the limitations and current constraints, and we are committed to further improving the diagnostic process with additional MRI data in the future. Thank you again for your valuable feedback.

Reviewer Comment 3.2

Ultrasound imaging is highly operator-dependent, and image quality varies significantly based on the performer, machine settings, and scanning conditions. Despite this, the study provides limited details on how ultrasound scans were conducted. For example, the probe is only classified as linear vs. curved, without specifying frequency settings or additional technical parameters that influence image

resolution.

Even in the prospective dataset collection (e.g., blind sweeping method), critical details are missing: Who performed the ultrasound? What equipment was used? (Machine models, manufacturers, and software versions.) Which probe settings were applied? (Frequency, gain, depth settings.)

Response:

We appreciate the reviewer’s comments regarding the operator-dependent nature of ultrasound imaging and the variability in image quality. Our original intention was to develop a highly robust and generalizable AI-assisted cranial diagnostic system. Therefore, in our initial prospective study, we did not specify detailed parameter settings. The case data we collected were derived from routine clinical cases in hospital settings. The parameter settings used across different hospitals adhered to standard guidelines.

To address the review’s concern, we have supplemented our original statistical analysis with more detailed statistics (see Supplementary Table 11). The table presents the mechanical index (MI), thermal Index (TIs), depth of X and Y, linear probe frequency, and curved probe frequency, with all numerical data displayed in mean (std) format. For MI and TIs, we used OCR technology to extract data from ultrasound images. Other data were retrieved from DICOM files. The abbreviations CS, SC, and PY represent the three participating centers. These additional statistics provide a more comprehensive understanding of the ultrasound imaging data, enhancing the clarity and robustness of our analysis.

Set	Development set		Internal test set	External test set	Blind sweeping set		
Subset	Train	Test		CS	SC	PY	
MI	1.17 (0.21)	1.15 (0.24)	1.28 (0.14)	0.87 (0.07)	0.89 (0.39)	0.96 (0.17)	1.37 (0.91)
TIs	1.09 (0.41)	1.09 (0.42)	1.24 (0.28)	0.64 (0.08)	0.19 (0.10)	0.19 (0.09)	0.97 (0.46)
Depth X (cm)	16.43 (4.15)	16.54 (4.44)	14.63 (4.78)	14.89 (0.98)	14.27 (1.25)	14.75 (1.25)	19.50 (2.52)
Depth Y (cm)	9.44 (2.42)	9.58 (2.58)	8.52 (2.64)	10.94 (0.45)	10.26 (0.69)	10.26 (0.69)	11.18 (2.16)
Linear (MHz)	9.32 (2.09)	8.96 (1.73)	9.17 (3.04)	8.71 (0.69)	8.52 (1.08)	8.40 (1.18)	7.550 (2.21)
Curved (MHz)	4.66 (1.17)	4.39 (1.27)	4.14 (1.42)	4.38 (1.17)	5.02 (0.97)	4.68 (1.12)	3.50 (1.09)

Mean (SD) values for Mechanical Index (MI), Thermal Index (TIs), depth (X and Y, cm), and probe frequencies (linear and curved, MHz) across development, internal, external, and blind

sweeping sets. CS, SC, PY represent Changsha, Sichuan and Guangzhou Maternal and Child Health Hospitals, respectively.

We also added a detailed table of manufacturer and model name (see Supplementary Table 2). Due to the table's extensive length, we have not presented it here. From the table, we can see that the data collected includes multiple manufacturers, such as GE Healthcare, Mindray, EDAN Instruments, Wisonic, and others, along with various models. This highlights the diversity of the data, which supports the generalizability of the collected dataset. It aligns with our original goal—to create a robust and versatile AI-assisted diagnostic system.

We added on page 3 (see Result section, paragraph 1):

“Table 1 provides a summary of the demographics and US scanners, and Supplementary Figure 1 shows the distribution of different disease, GA groups and probe types in the datasets. Supplementary Table 1 and 2 inform more information regarding imaging details, including transducer frequency, manufacturer, model name, depth and more.”

We also added a description of the personnel allocation for the blind sweeping validation on page 4 (see Result section, paragraph 2):

“Finally, radiologists with different levels (from interns to junior radiologists) collected blind sweeping data in rotation according to their daily duty schedules.”

We hope the added details address the reviewer’s concerns. We sincerely thank the reviewer for their valuable feedback.

Reviewer Comment 3.2

The study categorizes findings into "severe" and "mild" lesions, but this binary classification oversimplifies the clinical significance of intermediate cases. Grade I and II IVH are classified as mild, but these cases carry a risk of progression to higher-grade IVH, particularly in preterm infants. A three-tier classification—"normal," "mild/moderate," and "severe"—would be more clinically useful and practical.

Response:

We sincerely thank the reviewer for the insightful comment, and we fully agree that the binary classification may oversimplify the clinical significance of intermediate cases. We categorized Grade I or II IVH, ventricular dilatation and endymal cysts

into the mild class, and retrained the model to evaluate its performance into the three-class classification task.

We added a result table in Supplementary material (see Supplementary Table 7):

Set	Development test set			Internal test set			External test set		
Class	Normal	Mild	Severe	Normal	Mild	Severe	Normal	Mild	Severe
SEN	0.955	0.463	0.853	0.971	0.120	0.857	0.988	0.161	0.846
SPE	0.773	0.946	0.964	0.350	0.966	0.940	0.478	0.993	0.942
PPV	0.914	0.641	0.744	0.611	0.714	0.522	0.803	0.875	0.537
NPV	0.872	0.894	0.982	0.919	0.605	0.989	0.947	0.785	0.987
F1	0.934	0.538	0.794	0.750	0.206	0.650	0.886	0.272	0.657
AUC	0.922	0.842	0.971	0.680	0.624	0.939	0.832	0.754	0.959

SEN, sensitivity; SPE, specificity; PPV, positive predicted value; NPV, negative predicted value; F1, F1-Score;

From the results, it is evident that the model performs well and demonstrates robustness in the normal and severe categories, but its performance in the mild class is not very good. We further analyzed the misclassified cases.

We added a table to show the distribution of misclassified cases (see Supplementary Table 8):

Set	Category	Predict to	Predict to
		Normal (N,%)	Severe (N,%)
Internal test set	Grade I-II IVH(n=34)	19 (46.34)	15 (36.59)
	Ependymal cyst(n=37)	37 (80.43)	0 (0.00)
	Dilatation(n=2)	1 (50.0)	1 (50.0)
External test set	Grade I-II IVH(n=62)	52 (77.61)	10 (14.93)
	Ependymal cyst(n=16)	16 (84.21)	0 (0.00)
	Dilatation(n=2)	2 (66.67)	0 (0.00)

The results are displayed in number (percentage) format, with the percentage indicating the rate of prediction errors among all cases for each disease type.

From the table, we can observe that the model tends to predict ependymal cysts as normal. We speculate that this is related to the size of the cysts, as the model struggles to capture smaller cysts. Additionally, the model also performs poorly in predicting cases of Grade I-II hemorrhage and lateral ventricular dilatation. The model does not perform well in the mild class, and we believe this may stem from inherent challenges in distinguishing mild cases from normal cases due to potential overlap in their

characteristics. Furthermore, the subjective nature of medical image annotation could also contribute to variability in the labeling of mild cases. It's worth noting that from a clinical perspective, mild IVH typically has a favorable prognosis (see Supplementary Figure 4).

Given these challenges and the clinical relevance of distinguishing severe cases, the focus of this paper is on a binary classification task (severe vs. non-severe). However, we recognize the importance of finer-grained classification and will dedicate future research efforts to explore more robust grading systems and advanced methodologies to improve the accuracy across all severity levels, including the mild class. We sincerely thank the reviewer for their valuable feedback and guidance.

Reviewer Comment 3.3

Including clinical context in the AI system would have been helpful. For example, incorporating the patient's gestational age could significantly impact the interpretation of ultrasound findings.

Response:

We fully agree that incorporating clinical context, such as the patient's gestational age, birth weight and Apgar scores, into the AI system would enhance the accuracy and clinical relevance of predictions. As noted in the limitations section of our study, "To address this issue, we plan to expand data collection and explore additional methods to improve screening effectiveness from a multimodal perspective, incorporating demographic metadata and doppler US images. Existing work has also demonstrated that metadata used in traditional machine learning can yield good results in predicting the severity of IVH and mortality rates [Error! Reference source not found., Error! Reference source not found.]."

However, integrating multimodal data (including both imaging and textual information) into the current framework would require substantial modifications to the existing model architecture, which we regretfully cannot accomplish within the short timeframe of this revision. Nevertheless, we have already initiated follow-up research on multimodal diagnostics, with plans to incorporate clinical information, spectral imaging and color Doppler ultrasound into the model in the near future.

Reviewer Comment 3.4

The Introduction states that the dataset consists of 8,757 neonatal CUS images, yet the Results section presents different case numbers for internal and external validation. Additionally, the Introduction implies this is a single-institution study, while the Results indicate data collection from multiple hospitals. These inconsistencies should be clarified.

Response:

We apologize for any confusion caused by our original description. We corrected our description of page 3 (see Introduction section, paragraph 4):

"To address this gap, we proposed a Neonatal Cerebral Lesions Screening system (NCLS) to identify neonates at high risk for severe cerebral lesions. The system includes a view extraction module to detect key anatomical structures and extract standard views from CUS videos and a diagnostic module that integrates multiple views from the same neonate to predict lesion severity. We trained and developed these modules using an internal development set consisting of 8,757 neonatal CUS images, retrospectively collected from a single hospital, corresponding to 1,518 cases. To evaluate the performance of NCLS, we employed two distinct test sets. The first was an internal video test set, comprising 199 cases prospectively collected from the same hospital. The second was an external video test set, which included 356 cases prospectively collected from three other centers, enabling us to assess the clinical potential and generalizability of NCLS."

We have revised the Introduction section to clarify that the 8,757 neonatal CUS images were retrospectively collected from a single hospital, corresponding to 1,518 cases. The internal test set, consisting of 199 cases, was from the same hospital, and the external test set, comprising 356 cases, was collected from three additional centers. We appreciate the reviewer's helpful comments.

Reviewer Comment 3.5

Some sentences need revision for grammatical accuracy. For example: "Since mild abnormalities..." or "Blind sweeping CUS videos were inputted to..."

Response:

Thank you. According to your suggestion, we corrected the above grammatical errors and made an effort to correct the spelling and grammar errors and polish the whole manuscript.

Reviewer Comment 3.6

Some figure descriptions are misleading or incorrect. For example: Supplementary Figure 1 is described as including a pie chart and violin plot, but these are not present in the figure.

Response:

Thank you for your reminder. We have carefully reviewed all figures and tables throughout the manuscript and revised them in accordance with the publication format requirements of Nature Communications.

Source Data

As per the submission guidelines, we have included a “Source Data” zip file containing the following Excel files with the data supporting the figures and tables in the manuscript:

- **Source_data_Blind_randomized_trial.xlsx**: Contains data for Table 3 and Extended Figure 3, including AI_label (AI data index among four datasets), selected_label (senior radiologist’s chosen data index), Doctor1-3_result and AI_result (diagnostic results of four datasets), and expert_result (senior radiologist’s diagnostic result for the case).
- **Source_data_result.xlsx**: Contains data for Table 2, Figures 2–4, Supplementary Tables 3–5, 7, 8, and 14, representing binary classification results of AI and all radiologists for internal test set, external test set, and blind sweeping set, as well as junior radiologists’ AI-assisted results, ternary classification results, and state-of-the-art comparison results.
- **Source_data_statistic.xlsx**: Contains data for Table 1, Supplementary Tables 1–2, and Supplementary Figure 1, providing statistical information for each case.
- **Source_data_Volpe_reclassification_table.xlsx**: Contains data for Supplementary Figures 3–4 and Supplementary Table 11, showing reclassification results of cases from the Papile grading system to the Volpe grading system.
- **Source_data_Qualitative_scoring_results.xlsx**: Contains data for Figure 4, representing senior radiologists’ scoring of extracted sections across four dimensions.

Each file is clearly labeled, and the data within corresponds to the specified figures and tables as outlined above. Source Data are provided with this paper, as stated in the “Data Availability” section of the manuscript.

Reference:

1. Xie, B. et al. Computer-aided diagnosis for fetal brain ultrasound images using deep convolutional neural networks. *Int. J. Comput. Assist. Radiol. Surg.* 15, 1303–1312 (2020).
2. Krishna, T. B. & Kokil, P. Standard fetal ultrasound plane classification based on stacked ensemble of deep learning models. *Expert Syst. Appl.* 238, 122153 (2024).

3. Ahmad, T. et al. Can deep learning classify cerebral ultrasound images for the detection of brain injury in very preterm infants? *Eur. Radiol.* 1–11 (2024).
4. He, K., Zhang, X., Ren, S. & Sun, J. Deep residual learning for image recognition. in *Proceedings of the IEEE conference on computer vision and pattern recognition* 770–778 (2016).
5. Dosovitskiy, A. et al. An image is worth 16x16 words: Transformers for image recognition at scale. *ArXiv Prepr. ArXiv201011929* (2020).
6. Yu, W. & Wang, X. Mambaout: Do we really need mamba for vision? *ArXiv Prepr. ArXiv240507992* (2024).
7. Tschannen, M. et al. Siglip 2: Multilingual vision-language encoders with improved semantic understanding, localization, and dense features. *ArXiv Prepr. ArXiv250214786* (2025).
8. Manzari, O. N., Ahmadabadi, H., Kashiani, H., Shokouhi, S. B. & Ayatollahi, A. MedViT: a robust vision transformer for generalized medical image classification. *Comput. Biol. Med.* 157, 106791 (2023).
9. Yue, Y. & Li, Z. Medmamba: Vision mamba for medical image classification. *ArXiv Prepr. ArXiv240303849* (2024).
10. Liu, S., Zhang, L., Yang, X., Su, H. & Zhu, J. Query2label: A simple transformer way to multi-label classification. *ArXiv Prepr. ArXiv210710834* (2021).
11. Zhang, Z. et al. Finer-CAM: Spotting the Difference Reveals Finer Details for Visual Explanation. *ArXiv Prepr. ArXiv250111309* (2025).

We hope our revised manuscript will meet the Publication standards of your Journal Nature Communications. If you have any question about our manuscript, please don't hesitate to contact me.

Best Regards,

Luyao Zhou

Thanks for your kind work and consideration on publication of our manuscript entitled as "Deep learning approach for screening neonatal cerebral lesions on ultrasound in China " (manuscript NCOMMS-24-82232). On behalf of my co-authors, we would like to express our great appreciation to the reviewers. We have studied reviewers' comments carefully and made some corrections.

As follows are the point-by-point responses to the reviewers' comments.

Reviewer 1

The manuscript is much better in its revised form. All my comments and remarks have been addressed. I recommend the acceptance of the paper.

Response:

Thank you very much.

Reviewer 2

The authors have responded more than satisfactorily to my comments, including in their analysis not only the old Papile classification but also the more modern Volpe classification. I have no further questions or comments. I have not reviewed the code.

Response:

We greatly appreciate your thoughtful review and are pleased that our revisions have met your expectations.

Reviewer 3

Reviewer comment 3.1

1. Thank you for including technical details regarding the ultrasound acquisition protocols. However, a critical component that appears to be missing is information about the individuals who performed the ultrasound examinations. Specifically, it would be helpful to clarify whether sonographers, radiologists, or trainees performed the scans, and what level of training or experience they had prior to image acquisition. If these data are unavailable, I recommend addressing this limitation in the Discussion section.

Response

We thank the reviewer for the important concern regarding the demographics of the ultrasound data collection personnel. All cranial US images were acquired by

sonographers, junior radiologists, or trainees from multi-center. However, we are not able to provide the details of these personnel data. We have added this limitation in Discussion section.

“Last but not least, this study did not reveal detailed information about the individuals who acquired the CUS images. Variations in operator experience could impact the generalizability of the NCLS.”

We have added information about the image acquisition personnel (see the "Data Collection and Preprocessing" section in Methods):

“All CUS data were acquired by sonographers, junior radiologists or trainees with 1 to 3 years of experience of pediatric ultrasound and stored in Digital Imaging and Communications in Medicine (DICOM) format.”

Reviewer Comment 3.2

The manuscript suggests that the observed heterogeneity in ultrasound data supports the generalizability of the AI model. While this is a potentially valuable observation, it would strengthen the manuscript to elaborate on whether any measures were taken to ensure or enhance generalizability during model development.

Response

We thank the reviewer for the valuable comment. In our original submission, information regarding the strategies to enhance model generalizability was dispersed across multiple sections. To improve clarity and coherence on this point, we have provided a brief summary of the strategies used to enhance model generalizability as described in the manuscript.

1. Data diversity: Training data were collected using an inclusive strategy across a wide range of scanners, gestational ages, and clinical conditions, minimizing selection bias and promoting broad coverage of real-world variability.

2. Training augmentations: We applied RegMixup for the diagnosis task and mosaic augmentation for the detection task, both of which are known to improve model robustness by increasing input diversity and reducing overfitting. During inference, Test-Time Augmentation was used to improve prediction stability and calibration. These strategies were empirically validated through ablation studies during model development, confirming their positive impact on performance.

3. Evaluation and visualization: The model was tested on internal, external, and blinded datasets. Consistently high performance across these datasets demonstrates strong generalization. And we use Finer-CAM and verified that the model consistently attended to clinically meaningful anatomical regions, regardless of image quality or scanner differences, supporting its reliance on generalizable features.

We hope that the above summary clearly conveys our efforts and enables both reviewers and readers to appreciate the measures we took to ensure the model's generalizability.

Best regards,

Luyao Zhou